# SELF-IMPROVING LOOPS
# FOR VISUAL ROBOTIC PLANNING

**Calvin Luo**[*,1] , **Zilai Zeng**[*,1], **Mingxi Jia**[1], **Yilun Du**[2], **Chen Sun**[1]
[1]Brown University, [2]Harvard University

## ABSTRACT

Video generative models trained on expert demonstrations have been utilized as performant text-conditioned visual planners for solving robotic tasks. However, generalization to unseen tasks remains a challenge. Whereas improved generalization may be facilitated by leveraging learned prior knowledge from additional pre-collected offline data sources, such as web-scale video datasets, in the era of experience we aim to design agents that can continuously improve in an online manner from self-collected behaviors. In this work we thus propose the Self-Improving Loops for Visual Robotic Planning (SILVR), where an in-domain video model iteratively updates itself on self-produced trajectories, and steadily improves its performance for a specified task of interest. We apply SILVR to a diverse suite of MetaWorld tasks, as well as two manipulation tasks on a real robot arm, and find that performance improvements continuously emerge over multiple iterations for novel tasks unseen during initial in-domain video model training. We demonstrate that SILVR is robust in the absence of human-provided ground-truth reward functions or expert-quality demonstrations, and is preferable to alternate approaches that utilize online experience in terms of performance and sample efficiency. Visualizations and code are provided at diffusion-supervision.github.io/silvr/.

## 1 INTRODUCTION

Advancements in video generative modeling capabilities have directly led to their increased utilization as visual planners for robotic applications (Du et al., 2024b; Yang et al., 2023b; Ko et al., 2024; Liang et al., 2024). The synthesized visual plan, in the form of video frames generated with text conditioning, can be translated into executable actions via inverse dynamics models (IDMs). Intuitively, the data on which the video generative models and the IDMs are trained can greatly impact robotic performance and generalization. When explicitly optimized on in-domain examples of expert behavior, such visual planners are able to synthesize successful plans for solving demonstrated tasks in a robust manner. However, for arbitrary robotic behaviors, expert-quality demonstrations may not be readily available, and collection may be prohibitively expensive. It is thus worthwhile to investigate how visual planning can automatically *adapt* and *generalize* to novel tasks of interest.

Recent work has investigated how base generalization performance for visual planning can be improved by integrating knowledge from large-scale datasets of text and video collected from the internet. Adapt2Act (Luo et al., 2025) creates a powerful, generalizable, text-conditioned visual planner by combining a large-scale model pretrained on web-scale video data with a video model trained on a small set of in-domain demonstrations via score composition. At a high level, the adapted video model draws upon large-scale motion priors and powerful zero-shot text conditioning capabilities from the web-pretrained video model to facilitate generalization. Simultaneously, it can leverage the in-domain video model to better generate visual plans that respect the environment-specific visual characteristics and dynamics of the robotic setting. The result is an adapted video model that can generate in-domain-appearing plans for novel, unseen tasks conditioned on natural language.

Despite extending the amount of data utilized for visual planning to internet-level, the model still only has access to purely offline data. In the era of experience, we aim to design agents that can continuously improve from self-collected behaviors and feedback. In such a way, the agent can break free beyond the limits of offline data and learn by itself to refine performance on a specified task

---

*: Equal contribution. Correspondence to: calvin_luo@brown.edu and zilai_zeng@brown.edu.

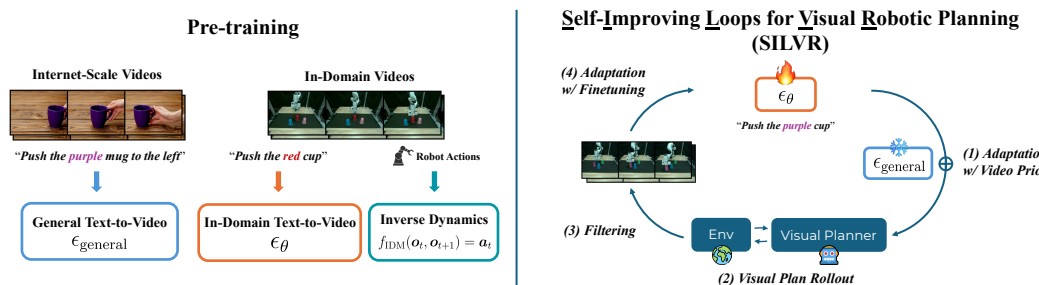

Figure 1: **SILVR Framework.** SILVR has access to two pretrained video generative models (left): one pretrained generally on internet-scale data and another pretrained on a general set of in-domain demonstrations. By default, SILVR uses the in-domain video model as a visual planner, which when utilized to interact with the environment, is able to achieve successful trajectories even for initially unseen tasks. These trajectories are then iteratively fed back to finetune the in-domain model (right), thus improving the overall quality of future visual planning as a whole through self-collected online experience. SILVR can optionally incorporate internet-scale pretrained video models as prior, which particularly improves performance in the case of real-world robotic experiments.

of interest. We therefore propose **Self-Improving Loops for Visual Robotic Planning** (**SILVR**), where the generated video iteratively self-improves with online experience, particularly with respect to behaviors previously unseen in the initial dataset of environment demonstrations. As shown in Figure 1, each loop adapts the video generative model (and optionally, the IDM) with environment-grounded data collected by the robotic agents following their own-generated visual plans.

SILVR utilizes a sparse reward signal to filter online experience for further finetuning of the visual planner; however, it is quite natural to consider alternative methods beyond visual planning or direct finetuning of the generative model. In our experiments we demonstrate that visual planning is superior to direct behavior cloning in both initial generalization performance as well as self-improvement capability, given the same amount of offline data and online experience. Furthermore, the final resulting visual planner can be distilled into a lightweight policy via behavior cloning for both fast and performant decision-making. We further showcase how SILVR is more sample efficient than reinforcement learning finetuning approaches, making it more applicable to real-world robotic settings. SILVR is also robust to the quality of the sparse reward signal; rather than requiring a human-defined ground-truth success function, we demonstrate that iterative improvements still arise when utilizing a pretrained vision-language model (VLM) to score experience based on the task descriptions.

To summarize, SILVR enables superior sample-efficient self-improvement over initially unseen tasks through visual planning, naturally integrating internet-scale pretrained video priors over text-alignment and motion when necessary, in comparison to regular action-prediction policies. During deployment, a policy can be "distilled" from SILVR's components for fast inference, showcasing how SILVR facilitates effective generalization and self-improvement for robotic tasks without sacrificing final execution speed.

We perform extensive evaluations of SILVR on the MetaWorld task suite, focusing on novel tasks unseen during initial training of the in-domain video model. We discover that the success rate of following visual plans synthesized through SILVR indeed continuously improves, by as much as **285%** over 10 iterations. We also apply SILVR to a real-world robot arm for two distinct manipulation tasks: selecting and pushing a colored object, and selecting and opening a colored drawer. We show how SILVR also naturally enables the incorporation of priors from internet-scale pretrained video generative models to facilitate task and visual generalization in real-world visual settings and dynamics. We demonstrate that performance for color combinations unseen during the initial offline training improves over multiple iterations through SILVR.

## 2  RELATED WORK

**Video Generation for Decision Making.** Recent advances in video models have achieved unprecedented visual quality and physical fidelity for video synthesis (Guo et al., 2023; Yang et al., 2024b;

Brooks et al., 2024; Veo-Team et al., 2024; Wang et al., 2025). This has demonstrated promise in summarizing world dynamics through videos (Yang et al., 2024a; Bruce et al., 2024) and has inspired the application of video models to solving decision-making problems (Escontrela et al., 2023; Du et al., 2024a; Yang et al., 2023b; McCarthy et al., 2024; Liang et al., 2024). Prior works have utilized video generative models as reward functions (Luo et al., 2024; Escontrela et al., 2023; Huang et al., 2023), dynamics models (Yang et al., 2023b; Bruce et al., 2024; Valevski et al., 2024), and pixel-based planners (Ko et al., 2024; Ajay et al., 2023; Du et al., 2024b; Zhou et al., 2024). As in UniPi (Du et al., 2024b), we employ video models to predict text-conditioned visual plans that depict future outcomes, which are subsequently translated into actions via inverse dynamics. While the performance of such visual planners may often be limited by their offline pretraining data, our approach allows iterative improvement by learning from online environment interactions.

**Self-Improving Generative Models.** Continuously improving by learning from self-produced cumulative experience is an essential capability of intelligent agents. Prior work has demonstrated the effectiveness of improving LLMs with their self-generated outputs (Yu et al., 2024; Tian et al., 2024; Huang et al., 2022), where the LLM can serve as its own reward function (Yuan et al., 2024) for preference optimization or data synthesizer (Patel et al., 2024) for supervised finetuning. However, a similar self-improvement recipe for video generation models remains underexplored. Most relevant to our work, VideoAgent (Soni et al., 2024) refines video generation through self-conditioning consistency and feedback from a VLM, and collects the successful plan rollouts for finetuning. We instead base our improvement loop on self-adaptation, where we leverage internet-scale video priors to synthesize improved visual plans for tasks unseen during initial in-domain training. Furthermore, our approach can still achieve self-improvement even with an initial model trained on suboptimal data and a notable relaxation on filtering requirements for finetuning data.

**Reinforcement Learning Finetuning of Behavior Cloning Policies.** Behavior cloning approaches such as Diffusion Policy (Chi et al., 2023), which implements a policy as a diffusion model trained on offline-collected experience, are a performant approach for decision-making. There have been numerous approaches for finetuning pretrained diffusion policies with respect to online experience and rewards. ResIP (Ankile et al., 2025) utilizes a frozen diffusion policy model to propose action predictions, and learns a policy on top using reinforcement learning that transforms it into a more accurate action to perform in the environment. DPPO (Ren et al., 2024) treats the sampling procedure of a diffusion policy as an internal Markov Decision Process, and explicitly finetunes the weights of the pretrained diffusion policy with respect to achieved rewards. DSRL (Wagenmaker et al., 2025) utilizes a frozen diffusion policy in a deterministic fashion to map noise samples to action samples, and learns a noise selector through reinforcement learning. In this work, we show that SILVR is more sample efficient than reinforcement learning finetuning of behavior cloning policies, and can achieve faster iterative improvements with respect to online experience.

## 3 METHOD

We introduce the Self-Improving Loop for Visual Robotic Planning (SILVR), in which a video generative model initially trained on a general set of in-domain demonstrations iteratively improves its visual planning performance for a particular task of interest in a self-adaptive manner. In Section 3.2, we describe how a small in-domain video model can be integrated with a generally pretrained text-to-video model to produce a strong, generalizable in-domain visual planner for real-world visual settings. Finally, in Section 3.3, we demonstrate how SILVR bootstraps an in-domain video model into a high-performing visual planner for solving a novel robotic control task through iteratively fine-tuning on self-collected experience.

### 3.1 VIDEO MODELS AS VISUAL PLANNERS

Synthesizing a visual plan in imagination and then executing it by converting it into actions is an intuitive and effective way to utilize video generative models for decision making. Prior work has applied text-guided video generation successfully for task planning (Du et al., 2024a;b; Ajay et al., 2023; Luo et al., 2025), across a variety of robot configurations and environment settings.

Specifically, we base our implementation on the UniPi framework (Du et al., 2024a), in which a text-to-video model is used to synthesize a text-conditioned sequence of future frames as a task plan. To physically realize the plan, we use a separately trained inverse dynamics model (IDM) to

---

**Algorithm 1** Self-Improving Loops for Visual Robotic Planning (SILVR)

---

    **Input:** Initial in-domain video model $\epsilon_\theta$, Inverse dynamics model $f_{\text{IDM}}$, Frozen internet-pretrained video prior $\epsilon_{\text{general}}$, Number of iterations $K$, Number of rollouts per iteration $N$, Environment `env`, Task prompt $g$, In-domain initial training data $\mathcal{D}_{\text{ini}}$, Data filter $f_r$

    **Output:** Self-improved in-domain video model $\hat{\epsilon}_\theta$, Optional distilled policy $\pi$

---

1:  $\hat{\epsilon}_\theta \leftarrow \epsilon_\theta$
2:  $\mathcal{D} \leftarrow \mathcal{D}_{\text{ini}}$ or $\phi$                                        $\triangleright$ Initialize finetuning data with $\mathcal{D}_{\text{ini}}$ or an empty set
3:  **for** $i = 1, ..., K$ **do**
4:     $\mathcal{D}_{\text{self}} \leftarrow \phi$
5:     $\tilde{\epsilon} \leftarrow$ `Adaptation`$(\hat{\epsilon}_\theta, \epsilon_{\text{general}}, g)$
6:     **for** $j = 1, ..., N$ **do**
7:         `env.reset`$(g)$
8:         $\mathcal{D}_{\text{self}} \leftarrow \mathcal{D}_{\text{self}} \cup f_r(\text{`Visual\_Planning\_Rollout`}(\text{env}, \tilde{\epsilon}, f_{\text{IDM}}))$
9:     **end for**
10:    $\mathcal{D} \leftarrow \mathcal{D} \cup \mathcal{D}_{\text{self}}$
11:    Finetune in-domain model $\hat{\epsilon}_\theta$ with accumulated data $\mathcal{D}$   $\triangleright$ $f_{\text{IDM}}$ can be optionally finetuned
12: **end for**
13: Optionally distill $\hat{\epsilon}_\theta$ into a lightweight policy $\pi$
14: **return** $\hat{\epsilon}_\theta, \pi$

---

translate consecutive pairs of visual frames into executable robotic actions, which are then directly performed in interaction with the environment. Visual planning offers the practitioner flexible computational tradeoffs; at a high level, replanning often incurs high computational cost but generally increases accurate plan following, whereas replanning infrequently is cheap but may suffer from error compounding. In this work, we focus on how such a video generative model can generalize and self-adapt to a novel task of interest through online self-collected experience.

## 3.2 INVERSE PROBABILISTIC ADAPTATION

Prior work (Luo et al., 2025) has investigated how in-domain demonstration data can best be integrated with large-scale pretrained video models for generalizable visual planning; in this work we leverage similar insights to successfully integrate on-the-fly experience into visual planners for iterative self-improvement. Inverse Probabilistic Adaptation (Luo et al., 2025; Yang et al., 2023a) (IPA) is a training-free approach that adapts generally pretrained text-to-video models for domain-specific video generation. To perform adaptation, the score predicted by an in-domain video model $\epsilon_\theta$ trained on a small sample of demonstrations is composed with the score prediction of a web-scale pretrained model $\epsilon_{\text{general}}$ during the sampling procedure, as depicted in the function below:

$$\tilde{\epsilon}_{\text{inv}} = \epsilon_{\text{general}}(\tau_t, t) + \alpha\Big(\epsilon_{\text{general}}(\tau_t, t \mid \text{text}) + \gamma\epsilon_\theta(\tau_t, t \mid \text{text}) - \epsilon_{\text{general}}(\tau_t, t)\Big) \tag{1}$$

where $\gamma$ is the prior strength, and $\alpha$ is the guidance scale of text-conditioning. Intuitively, the small in-domain text-to-video model serves as a probabilistic knowledge prior that guides the generation process of the small in-domain model during sampling. Prior work (Luo et al., 2025) has found that a visual planner constructed through IPA exhibits both strong generalization capability and in-domain understanding; it is able to synthesize performant visual plans that appear in-domain even for novel tasks unseen during video model training. This may stem from the fact that IPA utilizes the large-scale pretrained model, which inherently has stronger text-conditioned generalization, as the main denoiser. While Luo et al. (2025) based their conclusions on experiments in simulated environments, we believe the true promise of web-scale pretrained models lies in their powerful prior for real-world generalization scenarios, as demonstrated by our Panda arm object manipulation evaluations.

## 3.3 SELF-IMPROVING LOOPS FOR VISUAL ROBOTIC PLANNING

For visual planning approaches, task performance is fundamentally a fixed function of the video models used, and by extension, the data observed. Even when utilizing IPA, which can improve text-conditioned generalization to novel tasks by effectively increasing the amount of offline data

utilized to internet-scale, performance is set after adaptation. As a result, in this paper, we wish to design agents that can not only leverage offline data as a helpful prior for generalization, but also extend beyond it to continuously improve from self-collected online experience data.

We therefore propose **Self-Improving Loops for Visual Robotic Planning (SILVR)**, a framework that combines offline data with online experience to create a visual planner that iteratively improves for a particular task of interest. SILVR is initialized with an in-domain video model $\epsilon_\theta$ pre-trained on a set of task demonstrations within the environment. In each iteration, the in-domain video model is optionally integrated with a large-scale pretrained video model $\epsilon_{\text{general}}$ through IPA. The video model is utilized as a visual planner to interact with the environment and solve tasks not necessarily observed in the initial training stage; in SILVR, the trajectories collected through this interaction are used for further finetuning of the in-domain video model (as shown in Algorithm 1). As the in-domain model adapts to its own self-collected experience from deployment on a novel task, it improves its ability to solve that particular task over time. In this way, SILVR iteratively bootstraps an in-domain video model into a strong visual planner for a novel task of interest through a self-adapting improvement cycle.

We demonstrate that the visual planning approach enables strong self-improvement in a virtuous loop in a sample efficient manner, compared to reinforcement learning finetuning of behavior cloning models. We further stress-test our framework through ablations, and demonstrate that ground-truth human-defined reward functions can be replaced with an automated VLM success signal without inhibiting iterative self-improvement from occurring, and that SILVR can handle suboptimal initial data quality We find that SILVR is a robust approach for iteratively adapting to a task through effective utilization of both offline data and online experience, and reduces requirements on human-supplied components both in terms of feedback and demonstration quality.

Whereas visual planning demonstrates strong self-improvement capabilities, and can flexibly integrate in benefits from large-scale pretrained video models, it can be slow in execution compared to direct policy approaches. However, after applying SILVR, the final video planning components can be distilled into a lightweight policy through behavior cloning for future inference. We demonstrate in our experiments that such a final distilled policy achieves higher performance than applying self-improvement techniques to such a policy from the start, and can even demonstrate slight improvement over the final performance of the visual planner teacher. Thus, SILVR enables a successful balance between visual planning for improved and sample-efficient utilization of online experience, and a lightweight distilled policy for reactive inference during deployment.

## 4  EXPERIMENTS

We investigate how SILVR can improve an in-domain video model initially trained on a limited set of demonstrations and tasks to further solve novel robotic control tasks through self-collected experience. We focus on two main robot settings to evaluate SILVR: the MetaWorld-v2 (Yu et al., 2020) simulated environment, and a real-world Franka Emika Panda robot arm. We describe our experimental setup for each environment, as well as different design decisions considered.

### 4.1  EXPERIMENTAL SETUP AND EVALUATION

**Synthetic Environment:** MetaWorld encompasses a wide selection of tasks, allowing us to thoroughly assess visual planning performance trends through SILVR for many choices of held-out novel tasks. Furthermore, MetaWorld provides ground-truth success evaluations, enabling strictly quantitative comparisons on task performance as well as iterative improvement abilities. For MetaWorld experiments, we first collect 25 demonstrations from 8 different tasks (denoted with an asterisk in Table A2) for initial in-domain video model and inverse dynamics model training. Subsequently, we instantiate SILVR with both models on 12 unseen tasks for self-improvement. In each SILVR iteration, we collect 30 trajectories rendered from the environment via visual planning, and finetune both in-domain video model and inverse dynamics model using filtered successful data. Due to the sim-to-real gap, we disable the use of internet-scale video prior in MetaWorld by default.

**Real-World Environment:** Deploying SILVR on a robot arm in the real world demonstrates the practicality of the approach, as well as tests robustness to real-world confounding factors such as lighting conditions. In one experiment, we utilize a Franka Emika Panda robot arm for the task

| Iteration | 0 | 1 | 2 | 3 | 4 |
|---|---|---|---|---|---|
| DSRL (w/ GT Filter) | 9.4 (±1.7) | 8.3 (±1.6) | 7.4 (±0.9) | 7.5 (±3.4) | 7.7 (±3.4) |
| BCIL (w/ GT Filter) | 5.6 (±0.6) | 12.3 (±2.3) | 20.9 (±1.6) | 23.3 (±0.4) | 23.2 (±0.9) |
| SILVR (w/ GT Filter) | 14.7 (±0.6) | 27.7 (±1.9) | 33.5 (±2.2) | 43.5 (±2.6) | 44.2 (±4.5) |
| SILVR (w/ VLM Filter) | 17.0 (±0.6) | 24.4 (±0.9) | 28.7 (±2.8) | 34.4 (±1.4) | 38.4 (±1.3) |
| SILVR-Distilled DP | | | | | 44.2 (±4.5) → 49.2 (±3.4) |

Table 1: **SILVR Results on MetaWorld**. We report the average performance over 12 unseen Meta-World tasks for SILVR and all baseline methods, each aggregated over three seeds. We also provide the performance of diffusion policy distilled from the video model from the last SILVR iteration, denoted as "SILVR-Distilled DP". SILVR outperforms all baselines by a large margin since Iteration 1. Furthermore, SILVR-Distilled DP achieves the best overall performance.

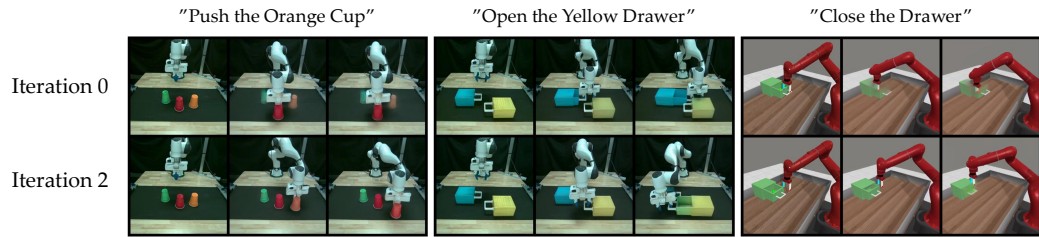

Figure 2: **Qualitative results on visual plans improvement.** We illustrate visual plans for a variety of tasks and settings at Iteration 0 (top) and Iteration 2 (bottom) with random initial object locations. Although the visual plan at Iteration 0 renders blurry objects and fails to complete the specified tasks, our approach synthesizes the correct visual plan (with slight color drift) after two SILVR iterations.

of pushing cups specified by a user-provided text prompt. In contrast to the MetaWorld setups, where each task of interest has its own distinct visual setting, we construct the cup experiment as a consistent scene setting of 3 differently colored cups (Figure 1). Success is then measured in terms of whether the robot arm can accurately locate a specified color cup and push it forward. To test generalization, conditioned on natural language, we evaluate successful planning and execution performance on unseen cup colors. In practice, we use a set of four colors (red, green, blue, pink) for in-domain training and two novel colors for testing generalization (orange, purple). This translates to 12 possible unique tasks formed from combinations of the seen colors, and we train our in-domain video model with 10 human-teleoperated demonstrations of each for a total of 120 training videos. Then, generalization evaluation is calculated as an average over 5 rollouts for every possible pair combination of the seen color set combined with the novel color, for a total of 30 videos. For both novel colors, we initialize SILVR using the same pretrained in-domain video model. In each SILVR iteration, we combine self-collected data with the initial demonstrations for in-domain finetuning.

In a second real-robot experiment, we utilize the Panda arm to select and open a drawer specified via a user-provided text prompt. The scene is constructed as two distinctly colored closed drawers, where the robot is prompted with one particular color and expected to open its corresponding drawer. We use a set of three colors (red, green, blue) for in-domain training and one novel color (yellow) for testing generalization. With 24 possible drawer placement combinations for each ordered pair of seen colors, of which there are six, this amounts to a total of 144 human-teleoperated demonstration training videos. Consistent with the cup pushing experiment, we use half the possible combinations for evaluation; therefore, performance is calculated as an average over 12 rollouts for every possible pairing of the novel color with a seen color, for a total of 36 self-collected trajectories per iteration.

For both real-robot experiments, success is judged by a human for evaluation. The same success signal is also used to perform data filtering on the rollouts. We study the impact of data filtering in Section 4.5, and enable adaptation with internet video priors through IPA by default.

**Implementation Details:** We implement our in-domain video model based on AVDC (Ko et al., 2024), with an added cross-attention layer to each level of the denoising U-Net to further improve text-conditioning capabilities. We train in-domain video models to predict 8 future frames condi-

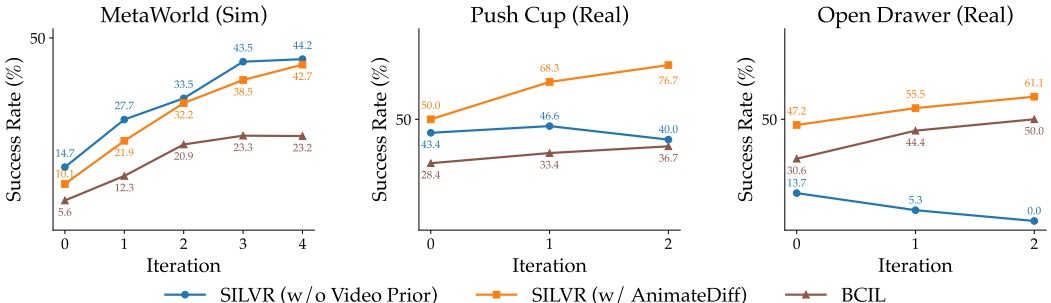

Figure 3: **SILVR Results** in comparison to Behavior Cloning Improvement Loop (BCIL). We report the average performance over 12 unseen MetaWorld tasks, as well as novel pushing and drawer opening tasks for Panda arm experiments across several iterations of self-improvement (x-axis). Numbers in the graph correspond to success rate achieved (y-axis).

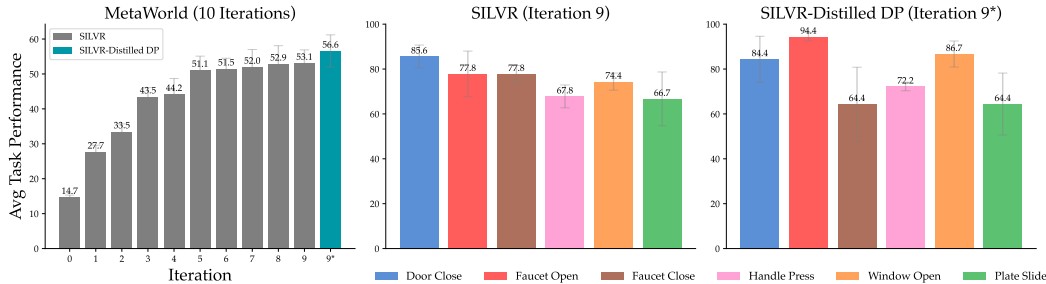

Figure 4: **SILVR results on MetaWorld for 10 iterations**. We report effects of training SILVR on an extended amount of iterations. On the left plot, we show that performance continues to monotonically increase, but with diminishing improvements and effective saturation past iteration 5. On the middle and right plots we visualize a comparison between the final iteration visual planner against its distilled student BC policy from the visual planner across 6 tasks, where we observe that certain tasks actually improve after distillation.

tioned on the current observation and task prompt, with a frame skip of 1 for MetaWorld and 16 for real-robot experiments. For the large-scale pretrained text-to-video model, we use AnimateD-iff (Guo et al., 2023) (~2B parameters), which is pretrained on WebVid-10M (Bain et al., 2021). Each iteration of SILVR finetunes the in-domain video model for 10,000 steps with a learning rate of 1e-6 on MetaWorld and 1e-5 on Panda Arm drawer opening tasks, and 8,000 steps with a learning rate of 2e-5 on Panda Arm pushing tasks. We investigate two IDM implementations; one that follows prior work (Luo et al., 2025) that implements the IDM as a MLP network that takes in the outputs of a VC-1 (Majumdar et al., 2023) encoder model, finetuned on in-domain demonstrations. Whereas the MLP-IDM was sufficient for real-world experiments, we found improved performance when using a Diffusion-IDM (DIDM), which is implemented as a Diffusion Policy (Chi et al., 2023) with an additional goal frame provided as conditioning, for simulated settings. Additional details on IDM design and hyperparameters are provided in Section B. For MetaWorld experiments, the DIDM is iteratively finetuned on the online collected experience along with the video model.

## 4.2 SILVR VIA FILTERED FINETUNING

We report incremental visual planning results for MetaWorld through five SILVR iterations, against two self-improving baseline methods built off of Diffusion Policy (DP) (Chi et al., 2023): DSRL (Wagenmaker et al., 2025) and Behavior Cloning Improvement Loop (denoted as "BCIL"). We initialize DSRL and BCIL with a Diffusion Policy trained on the same in-domain data as used for in-domain video model and inverse dynamics model pre-training in SILVR. We utilize the ground-truth task success signal to filter the same amount of self-collected data per iteration and finetune models with only successful trajectories.

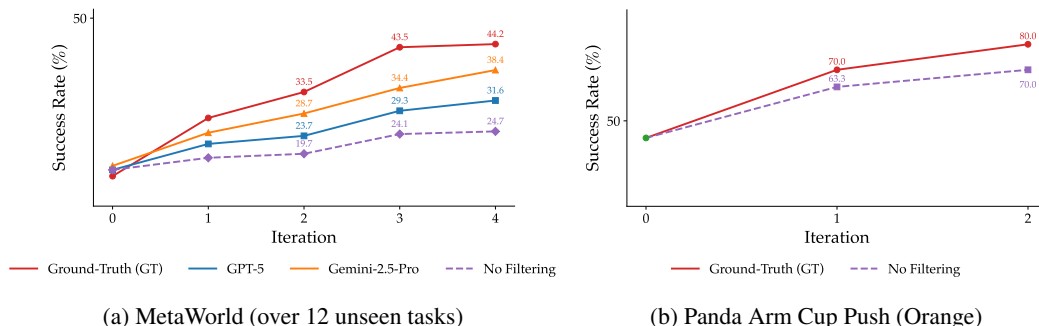

(a) MetaWorld (over 12 unseen tasks)  (b) Panda Arm Cup Push (Orange)

Figure 5: **Ablations on data filtering.** We compare the effect filtering has on success rate (y-axis) across iterations of finetuning (x-axis), on both MetaWorld (5a) and Panda arm (5b) setups. On MetaWorld (left plot), we further report accuracy when filtering is performed by a VLM. We observe SILVR consistently improves task even without access to ground-truth filtering signals.

In Table 1, the performance is averaged over 12 unseen MetaWorld tasks and aggregated over 3 seeds. Despite the same initial in-domain data being shared across all methods, we observe that the visual planning approach achieves better performance than DP-based approaches at Iteration 0, laying solid foundations for subsequent improvement. Compared to DP, which learns to map observations to actions directly, visual planning decouples dynamics modeling from action prediction. We hypothesize that the separately learned environment visual dynamics is easier to transfer when solving a novel task, leading to stronger base generalization performance through visual planning. While DSRL fails to improve and BCIL quickly saturates at a low success rate after several iterations, SILVR continuously improves and consistently outperforms the baselines by a large margin, demonstrating its high sample efficiency.

### 4.3 SILVR with Internet-Scale Video Prior

One important observation from Table 1 is that the generalization capability of the visual planner in solving novel tasks has a fundamental impact on self-improving dynamics. When the task of interest is exceptionally challenging or involves confounding factors, the capacity of the in-domain video model alone might not be sufficient to elicit self-improving behaviors. In the real-world experimental setup, we adapt our in-domain model with an internet-pretrained video prior, AnimateDiff, to further strengthen the zero-shot generalization and adaptability of the visual planner. In Figure 2, we qualitatively illustrate the improvements of generated visual plans after two SILVR iterations in combination with AnimateDiff. Without observing any demonstrations of the specified tasks at Iteration 0, the visual planner can synthesize plans with blurry objects where the robot arm executes the task incorrectly. On the other hand, two iterations of SILVR not only improve the clarity of the visual plans, but also demonstrate successful task completion behaviors in the same initial layout.

In the middle plot of Figure 3, we report the average performance of pushing cups in two unseen colors, orange and purple, aggregated over 30 rollouts per iteration. We discover that SILVR consistently improves over iterations when adapting with AnimateDiff. In the rightmost plot of Figure 3, we provide the SILVR results across iterations on opening a novel colored drawer, averaged over 36 rollouts per iteration, and find that SILVR bootstraps initial visual planning performance with the help of the internet video prior. However, in both real-world experiments, the visual planner struggles to improve or even continuously deteriorates without video prior. This highlights the importance of internet video prior in self-improving behavior under real-world setups with increased visual complexity and task difficulty. In simulated environments like MetaWorld, we observe that the self-improving trend occurs regardless of whether internet video priors are utilized or not, indicating that the benefits of utilizing internet priors may diminish when there is a substantial sim-to-real gap.

Consistent with our findings on MetaWorld, action-predictive behavior cloning has a lower base generalization performance and slower self-improvement trend compared with SILVR on real robots, as shown in the BCIL curves of the middle and right plots in Figure 3. This highlights a key benefit of SILVR: its ability to seamlessly utilize internet-pretrained video prior for improved text-conditioned generalization and sample-efficient online improvement in real-world robotic settings.

## 4.4 SILVR SATURATION AND DISTILLATION

To further understand the limits of self-improving behavior over iterations, we provide MetaWorld results for 10 SILVR iterations in Figure 4. We find that SILVR saturates at Iteration 5 with marginal gains in the following iterations. We hypothesize that this may potentially arise from discovered local minima in task-specific strategy, where similar experiences are collected until saturation. We believe that a possible mitigation is to introduce the notion of "exploration" into the visual planning framework to avoid "unimodal" behavior. Such research may look into how to extract out more diverse plans from the video planner by exploiting the stochastic nature of visual generative models. We leave this investigation as promising future work.

While visual planning approaches demonstrate strong performance in task generalization, their inference speed is bottlenecked by the video generation process, which can be prohibitively expensive for downstream applications. To mitigate this, we distill the video model from the last SILVR iteration into a lightweight diffusion policy. As shown in Figure 4, the SILVR-distilled diffusion policy at Iteration 9 significantly outperforms the BCIL baseline and achieves the best overall performance. A slight performance increase after distillation is a trend consistently observed across iterations, such as demonstrated in Iteration 4 of Table 1. This further demonstrates that SILVR not only excels in sample-efficient task adaptation, but also supports high inference efficiency for downstream deployment via distillation.

## 4.5 IMPACT OF DATA FILTERING SIGNALS ON SILVR

While utilizing self-collected data is a promising approach for scalable self-improvement, filtering collected experience often requires some level of human intervention, whether through manually determining successful trajectories or designing a heuristic for quality control. We therefore investigate how different filtering techniques affect SILVR performance, or if SILVR is robust to such design decisions. For both MetaWorld and Panda Arm settings, we compare between using a ground-truth or human-evaluated notion of success to filter what trajectories the in-domain model is finetuned on, against not using any filtering at all and utilizing all achieved trajectories irregardless of outcome. While ground-truth success signals can often be inaccessible, we investigate whether the current state-of-the-art VLMs, GPT-5 and Gemini-2.5-Pro, can provide useful task success signals and serve as a robust alternative to ground-truth signals on MetaWorld.

In Figure 5a, we observe that both GPT-5 and Gemini-2.5-Pro can still enable self-improving behavior across SILVR iterations when serving as a task success judge, in which Gemini achieves the best performance among all VLM filters. We also discover that without any data filtering, the improvement over each SILVR iteration appears to be marginal compared to the filtered setup. On the other hand, in Figure 5b, for the Panda arm, we observe that no filtering still facilitates continuous improvement over every iteration through SILVR when adapted with the internet video prior. This is an encouraging finding, as it suggests that even for settings where manual curation of experience is expensive, self-improvement can still occur. We attribute this property to score composition (Luo et al., 2025; Yang et al., 2023a); suboptimal demonstrations can still communicate useful information to the in-domain model, such as visuals, valid motions, and interaction dynamics of the specific deployment environment that when combined with an internet-pretrained video model can result in a final composed output plan can be both performant as well as appearing in-domain. As such, the success rate may still improve from iteration to iteration without filtering, as the in-domain model improves its modeling of environment dynamics and visuals, even from suboptimal self-collected experience over iterations.

## 4.6 SILVR WITH SUBOPTIMAL DATA

Visual planners are usually trained explicitly on expert in-domain demonstrations, which communicate not only environment-specific visual characteristics, physics, and interaction dynamics to the generative model during optimization, but also a notion of success and optimal behavior. However, for arbitrary environments, such expert-quality in-domain data can be expensive to collect and curate at scale. On the other hand, suboptimal demonstration data, such as utilizing random actions during the collection procedure, may generally be cheaper to gather; however, training on a large dataset of low-quality data may not result in a performant visual planning model capable of generating plans worth following. A natural question is how robust SILVR is to initialization data, or whether a performant video planner can still be created when only suboptimal demonstrations are available.

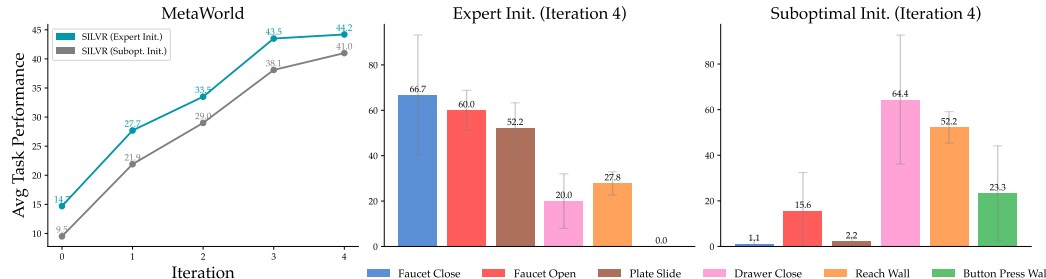

Figure 6: **Ablation studies on in-domain data quality**. On the left plot, we report SILVR performance averaged over 12 novel tasks and 3 seeds when the provided initial demonstrations of seen tasks are expert-quality or suboptimal. We find that SILVR successfully self-improves despite suboptimal data initialization. In the middle and rightmost plots, we visualize 6 tasks that have the most distinctive performance between expert and suboptimal SILVR initializations at the final iteration.

In our setting, we construct suboptimal data as simulated trajectories where 70% of the time a random action is selected and 30% an expert action is utilized. As a consequence of this interaction procedure, the resulting trajectories are unable to successfully solve complex tasks. In MetaWorld we find that SILVR still demonstrates continuously improving behavior when initialized with suboptimal data, as shown in the leftmost plot of Figure 6, highlighting the robustness of SILVR to initial data quality. Additionally, we select 6 tasks whose performance differs most between the expert and suboptimal setup, and report their performance on Iteration 4 in the middle and rightmost plots of Figure 6. We find that Faucet Open, Faucet Close and Plate Slide benefit most from initialization with expert demonstrations, indicating that the skills acquired from seen tasks can be essentially useful when solving these novel tasks. Meanwhile, Drawer Close, Reach Wall, Button Press Wall can benefit more from random exploration than expert actions from specific seen tasks, leading to stronger performance when the in-domain model is initially trained on suboptimal demonstrations.

## 5 CONCLUSION AND FUTURE WORK

**Conclusion.** We present SILVR, a self-improvement loop for solving novel robotic tasks via visual planning. Initialized from an in-domain video model pretrained on a small general set of demonstrations, SILVR iteratively updates the visual planner for a novel task of interest through self-collected online experience. Compared to equivalent behavior cloning setups, or utilizing online experience through reinforcement learning finetuning, we find that SILVR achieves superior self-improvement capabilities and demonstrates sample efficiency. We successfully apply SILVR as a self-improving visual planner not only for synthetic environments, but also on robot arms deployed in the real world.

**Limitations and Future Work.** SILVR implicitly assumes that the initial in-domain model, optionally adapted with an internet-pretrained video model, achieves a reasonable success rate to collect online experience and achieve self-improvement. This assumption may not hold when the novel task is too challenging. Additionally, the choice of internet-pretrained video model can pose a trade-off on between video quality and computation cost. Whereas in this work we choose AnimateDiff (Guo et al., 2023) as a large-scale pretrained video model with both reasonable generation quality as well as computational efficiency, more recent video generative models with enhanced visual quality can be further explored for improved visual planning performance on downstream robotic tasks.

Whereas SILVR achieves the best self-improvement performance with some initial success rate on the novel task, the cold start problem may pose challenges. Just as exploration can help address the cold start problem for standard reinforcement learning, we believe that investigating how to improve exploration for the visual planning framework in a principled manner is a promising future direction.

**Acknowledgments.** This work is partially supported by Samsung and the U.S. National Science Foundation under Cooperative Agreement No. 2433429. Our research was conducted using computational resources at the Center for Computation and Visualization at Brown University. We would like to thank Professors George Konidaris and Stefanie Tellex for their generous support for our real-robot experiments. We would also like to thank Shijie Wang and Zitian Tang for their help with the experiments. Calvin would like to thank Dennis Lee and Ally Koo, as well as Kayan Shih and her family, for all their kindness and hospitality during the paper writing process.

## 6 REPRODUCIBILITY STATEMENT

To support reproducibility, all codebases utilized in this paper were modified versions of publicly available repositories. Furthermore, we provide extensive details of hyperparameters and settings both in the main body of the text as well as in the Appendix.

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

## A    TASKS AND TEXT PROMPTS

Below we list the tasks and associated text prompts used for evaluating SILVR. Tasks with demonstrations seen during training of the in-domain model are denoted with an asterisk.

| Task | In-Domain Model Prompts | Internet-Domain Model Prompts |
|---|---|---|
| Coffee Button* | coffee button | a robot arm pressing the coffee machine button |
| Door Open* | door open | a robot arm reaching a door handle and pulling it to open the door |
| Drawer Open* | drawer open | a robot arm opening a drawer by pulling its white handle backward |
| Peg Unplug Side* | peg unplug side | a robot arm unplugging a peg by pulling it from the right to the left |
| Plate Slide Side* | plate slide side | a robot arm sliding a plate from the left into the net on the right side |
| Push* | push | a robot arm pushing an object forward to a green sphere |
| Reach* | reach | a robot arm reaching a red sphere |
| Sweep* | sweep | a robot arm moving an object to the left side of the table |
| Door Close | door close | a robot arm pushing a door to close it |
| Window Close | window close | a robot arm closing a window by pulling its handle from the left to the right |
| Window Open | window open | a robot arm opening a window by pushing its handle from the right to the left |
| Drawer Close | drawer close | a robot arm closing a drawer by pushing its white handle forward |
| Faucet Close | faucet close | a robot arm pulling a faucet counterclockwise |
| Faucet Open | faucet open | a robot arm pushing a faucet clockwise |
| Handle Press | handle press | a robot arm pressing down a handle |
| Handle Press Side | handle press side | a robot arm pressing down a handle on the side |
| Dial Turn | dial turn | a robot arm turning a dial counterclockwise |
| Plate Slide | plate slide | a robot arm sliding a plate forward into the net |
| Reach Wall | reach wall | a robot arm reaching toward a red sphere over a wall |
| Button Press Wall | button press wall | a robot arm reaching over a wall to press a button |
| Push Red Cup* | red | a robot arm pushing the red cup |
| Push Blue Cup* | blue | a robot arm pushing the blue cup |
| Push Green Cup* | green | a robot arm pushing the green cup |
| Push Pink Cup* | pink | a robot arm pushing the pink cup |
| Push Orange Cup | orange | a robot arm pushing the orange cup |
| Push Purple Cup | purple | a robot arm pushing the purple cup |
| Open Red Drawer* | red | a robot arm opening the red drawer |
| Open Green Drawer* | green | a robot arm opening the green drawer |
| Open Blue Drawer* | blue | a robot arm opening the blue drawer |
| Open Yellow Drawer | yellow | a robot arm opening the yellow drawer |

Table A2: **Task-Prompt Pairs.** We include a comprehensive list of tasks and their text prompts for in-domain training and evaluation. "∗" denotes tasks seen during initial training of the in-domain model. We also provide the prompts used to interface with the internet-pretrained text-to-video model during adaptation with IPA.

## B    IMPLEMENTATION DETAILS

We provide detailed architecture configurations of the models used in SILVR, and their relevant hyperparameter settings below.

**MLP Inverse Dynamics Model:** Following prior work (Luo et al., 2025), we design one choice of inverse dynamics model as a small MLP network built on top of a pretrained pixel-based representation network. The MLP-IDM takes as input the embeddings of two video frames, which are extracted using VC-1 (Majumdar et al., 2023), and outputs a prediction of the action that enables the transition between the provided frames.

For the Panda arm experiments, the MLP-IDM is tasked with predicting the end effector position of the last frame provided. This is then executed in the physical environment through inverse kinematics. Furthermore, the two video frames have a frameskip of 16; the frequency at which the camera is queried for trajectories is so high such that two temporally consecutive frames is not more substantially meaningful than just observing the last frame. For MetaWorld experiments, the two video frames are consecutive, and thus have a frameskip of 1.

The total parameter count of the MLP-IDM used in real-world experimentation is 85.81M. Of these, 85.80M parameters are inherited from VC-1 whereas our MLP-IDM design contributes only an additional 10759 parameters due to the additional MLP on top.

In our real-world experiments, we reuse the same MLP-IDM for all tasks within the same environments, and do not perform any finetuning during the SILVR iterations with subsequently self-

collected data. In such a way, the MLP-IDM is trained on a set of seen tasks, but applied to a novel task without further modification. The subsequent success on such novel tasks therefore highlights not only the robustness of the MLP-IDM learned, but also the visual quality of the synthesized visual plans. The detailed hyperparameters of MLP-IDM training are provided in Table A3.

| Hyperparameter | Value |
|---|---|
| Input Dimension | 1536 |
| Output Dimension (Panda) | 7 |
| Training Epochs | 20 |
| Learning Rate | 1e-5 |
| Optimizer | AdamW |

Table A3: **Hyperparameters of MLP Inverse Dynamics Model Training.** We list the relevant hyperparameters of training the MLP inverse dynamics model.

**Diffusion Inverse Dynamics Model (DIDM):** Whereas the MLP-IDM does continue to facilitate self-improvement for MetaWorld experiments through SILVR, we find that the most performant implementation for our simulated experiments was a Diffusion Inverse Dynamics Model (DIDM). The DIDM is built off the UNet implementation of a Diffusion Policy (Chi et al., 2023); it is modified to take in not only the current frame but also a frame 9 timesteps into the future and outputs an action chunks of size 8. We further implement the DIDM to operate directly in StableDiffusion latent space, where each frame is of dimension (64,64,4), rather than RGB space of (512,512,3) for further speed efficiency. The DIDM is initially trained on the 8 seen-task set (with 25 initial demonstrations per task) for 200 epochs, with a learning rate of 1e-4, and a batch size of 128. At each SILVR iteration on MetaWorld, the DIDM is further finetuned on the 30 collected demonstrations for 30 epochs, reusing a lr of 1e-4, and with a batch size of 30.

| Hyperparameter | Value |
|---|---|
| Input Dimension | 32768 |
| Output Dimension (MetaWorld) | 4 |
| Training Epochs | 200 |
| Learning Rate | 1e-4 |
| Optimizer | AdamW |

Table A4: **Hyperparameters of Diffusion Inverse Dynamics Model Training.** We list the relevant hyperparameters of training the diffusion inverse dynamics model.

**In-Domain Model:** We reuse the implementation of a small-scale diffusion model that conditions on both natural language and an initial pixel frame from (Ko et al., 2024). To improve text-conditioned capabilities of the model, we add an additional Cross-Attention layer to every level of the U-Net, which attends to the CLIP-encoded text prompt. Specifically, we instantiate UNet with 3 ResNet blocks for MetaWorld settings and 2 ResNet blocks for Panda arm tasks. We report the detailed list of model parameters in Table A5. In total, the in-domain model consists of 179.91M parameters for MetaWorld and 156.58M parameters for Real-World experiments. We perform initial in-domain training for 70K training steps on MetaWorld and 88K steps on Panda, with a batch size of 8 and a learning rate of 2e-5. In each SILVR iteration, we finetune the in-domain video model for 10K steps with with a batch size of 4 and a learning rate of 1e-6 on MetaWorld. On Panda Arm, we finetune for 8,000 steps with a batch size of 8 and a learning rate of 2e-5 on Cup Pushing and for 10,000 steps with a batch size of 8 and a learning rate of 1e-5 on Drawer Opening. All experiments are performed on a single NVIDIA A6000 or RTX3090 GPU.

**Internet-Domain Model:** Following Adapt2Act (Luo et al., 2025), we employ AnimateDiff (Guo et al., 2023) as the frozen internet-pretrained video model for inverse probabilistic adaptation. Additionally, we use SparseCtrl (Guo et al., 2024) to enable image-conditioned video generation. Model components and their parameter counts are listed in Table A6. In total, AnimateDiff consists of 2.005B parameters.

| Component | # Parameters (Millions) |
|---|---|
| U-Net (MetaWorld) | 116.71 |
| U-Net (Panda Arm) | 93.38 |
| Text Encoder (`openai/clip-vit-base-patch32`) | 63.2 |

Table A5: **In-Domain Model Components.** SILVR relies on a small in-domain text-to-video model, which we base our implementation off of prior work (Ko et al., 2024). We list the size of the components of the model architecture used.

| Component | # Parameters (Millions) |
|---|---|
| VAE (Encoder) | 34.16 |
| VAE (Decoder) | 49.49 |
| U-Net | 1302.16 |
| Text Encoder | 123.06 |
| ControlNet | 496.73 |

Table A6: **AnimateDiff Components.** IPA relies on a internet-scale text-to-video model; in this work we use AnimateDiff. We thus list the size of components of the AnimateDiff checkpoint used. The checkpoint is used purely for inference, and is not modified or updated in any way. Note that the VAE Decoder is not utilized in our framework.

**Visual Planning Hyperparameters:** In visual planning, we predict 8 future frames conditioned on the current observation and task prompt. We follow (Luo et al., 2025) to perform DDIM (Song et al., 2021) sampling for 25 steps to synthesize visual plans, in which the text-conditioning guidance scale is set to 2.5 for MetaWorld experiments and 7.0 for Panda Arm Pushing. We use 0.5 as the prior strength for inverse probabilistic adaptation.

**Choices of Control Loop:** Visual planning provides the user control over the quality of execution against the speed. In our experiments, each visual plan consists of 9 frames, including one current observation and eight future frames, and can be translated into 8 actions. By performing open-loop control, we execute all 8 actions from a single visual plan sequentially in the environment without any re-planning. While synthesizing a visual plan can often involve multiple sampling steps and thus be time-consuming, open-loop control greatly improves the interaction efficiency. However, since open-loop control does not adjust the control actions based on the feedback from the environment, the subsequent actions from the plan might become suboptimal to the latest states and cause error accumulation. To mitigate this issue, closed-loop control adjusts the action for every interaction step. Specifically, we execute only the first action from the plan, and perform re-planning based on the new observation received from the environment. Although this control style allows us to interact most reliably, it incurs a large computational overhead due to frequent re-planning. To balance the execution quality and efficiency, we can flexibly choose a control loop between the two extremes of open-loop and closed-loop. For example, we execute half of the plan (e.g. 4 actions) before re-planning, which we reference as semi-open-loop control.

To achieve the best execution speed, we employ open-loop control in Panda Arm Pushing and Drawer Opening tasks, in which we discover that visual plans can be performed decently, with negligible deviation in the real execution. For all MetaWorld experiments, we utilize semi-open-loop control to balance performance and efficiency.

**Behavior Cloning Improvement Loop**: A Diffusion Policy is initially trained on the same data as used in SILVR for 150 epochs with learning rate 1e-4 and batch size 64. For each iteration, we deploy the diffusion policy to interact with the environment and collect 30 task demonstrations. The policy will then be fine-tuned on the successful data filtered by ground-truth task success signals for 50 epochs with batch size 30 and learning rate 1e-4.

**SILVR-Distilled Diffusion Policy**: After applying SILVR, we optionally distill the visual planning components into a diffusion policy. This has the benefit of lightweight, fast inference for final deployment, while still leveraging the self-improving benefits of visual models during the training

process, thus balancing both worlds. The architecture of the DP is consistent with that used in the DSRL and BCIL experiments. During distillation, the visual planner teacher model first collects 120 demonstrations from the environment. Then, a DP is trained for 300 epochs with a batch size of 64 and a learning rate of 1e-4 on only these 120 demonstrations collected by the teacher.

In essence, SILVR can be thought of as composed of two systems. The slower system is the video planner approach, where the advantage appears in greater autonomous improvement capabilities. The faster system is the distilled diffusion policy, which although has weaker self-improvement, can capture the current performance of the slower system and be deployed with fast inference.

**DSRL Implementation**: We utilize the open-source implementation of DSRL (Wagenmaker et al., 2025) for our experiments. We maintain the vast majority of parameters from the default setting, but make sure to collect 30 demonstrations for each iteration as in SILVR. When optimizing over the collected experience, we utilize a comparable amount of gradient steps with the amount found in the default settings of DSRL; we use a batch size of 256 with 60000 gradient steps per update. We find that given the same amount of experience and a sufficiently large update budget, DSRL is unable to match the iterative performance improvements of SILVR. This highlights the sample inefficiency of reinforcement learning, which may need extra experience to successfully bootstrap a value function for the novel task of interest.

We performed a light ablation, comparing a roughly equivalent amount of gradient updates with SILVR (150 gradient steps per update) against the amount of updates used across multiple standard robotic tasks provided by the DSRL publicly available codebase, and found no significant difference in improvement trend. This suggests that the bottleneck for self-improvement through DSRL is not update computation, but experience collection; on the other hand, SILVR demonstrates significantly better sample efficiency for iterative self-improvement.

| Iteration | 0 | 1 | 2 | 3 | 4 |
|---|---|---|---|---|---|
| DSRL (150 Updates) | 10.1 (±0.1) | 8.9 (±0.6) | 8.6 (±1.0) | 9.4 (±0.1) | 8.3 (±0.2) |
| DSRL (60000 Updates) | 9.4 (±1.7) | 8.3 (±1.6) | 7.4 (±0.9) | 7.5 (±3.4) | 7.7 (±3.4) |

Table A7: **DSRL Performance Ablation over Different Update Rates.** We report mean success rate and standard deviation across three iterations for DSRL with different numbers of updates per iteration. We show that the bottleneck for DSRL is not gradient updates, but collected experience.

## C  METAWORLD TASK PERFORMANCE DECOMPOSITION

| Iteration | 0 | 1 | 2 | 3 | 4 | 5 | 6 | 7 | 8 | 9 |
|---|---|---|---|---|---|---|---|---|---|---|
| Button Press Wall | 0.0 (±0.0) | 0.0 (±0.0) | 0.0 (±0.0) | 0.0 (±0.0) | 0.0 (±0.0) | 0.0 (±0.0) | 0.0 (±0.0) | 0.0 (±0.0) | 0.0 (±0.0) | 0.0 (±0.0) |
| Dial Turn | 4.4 (±1.9) | 25.6 (±22.7) | 17.8 (±15.8) | 23.3 (±20.8) | 24.4 (±19.0) | 25.6 (±22.2) | 28.9 (±25.9) | 36.7 (±31.8) | 36.7 (±32.8) | 34.4 (±30.2) |
| Door Close | 35.6 (±6.9) | 57.8 (±26.9) | 58.9 (±25.2) | 74.4 (±15.8) | 72.2 (±18.4) | 80.0 (±13.3) | 81.1 (±11.7) | 83.3 (±8.8) | 78.9 (±12.6) | 85.6 (±5.1) |
| Drawer Close | 1.1 (±1.9) | 11.1 (±7.7) | 20.0 (±6.7) | 21.1 (±6.9) | 20.0 (±12.0) | 31.1 (±10.7) | 27.8 (±10.7) | 27.8 (±13.5) | 28.9 (±19.0) | 31.1 (±13.5) |
| Faucet Close | 17.8 (±6.9) | 27.8 (±25.2) | 41.1 (±20.1) | 50.0 (±24.0) | 66.7 (±26.5) | 67.8 (±13.9) | 73.3 (±16.7) | 80.0 (±13.3) | 73.3 (±17.3) | 77.8 (±3.8) |
| Faucet Open | 6.7 (±6.7) | 24.4 (±18.4) | 37.8 (±6.9) | 57.8 (±6.9) | 60.0 (±8.8) | 82.2 (±1.9) | 78.9 (±10.2) | 80.0 (±10.0) | 71.1 (±11.7) | 77.8 (±10.2) |
| Handle Press | 35.6 (±7.7) | 41.1 (±10.2) | 40.0 (±3.3) | 56.7 (±8.8) | 58.9 (±13.5) | 56.7 (±3.3) | 63.3 (±5.8) | 70.0 (±5.8) | 72.2 (±9.6) | 67.8 (±5.1) |
| Handle PressSide | 18.9 (±5.1) | 31.1 (±12.6) | 48.9 (±19.0) | 72.2 (±13.5) | 60.0 (±8.8) | 70.0 (±14.5) | 75.6 (±8.4) | 66.7 (±6.7) | 68.9 (±18.4) | 65.6 (±11.7) |
| Plate Slide | 17.8 (±5.1) | 26.7 (±16.7) | 33.3 (±17.3) | 47.8 (±15.8) | 52.2 (±11.7) | 70.0 (±5.8) | 60.0 (±5.8) | 62.2 (±18.4) | 71.1 (±15.0) | 66.7 (±12.0) |
| Reach Wall | 10.0 (±5.8) | 32.2 (±1.9) | 22.2 (±9.6) | 30.0 (±6.7) | 27.8 (±5.1) | 35.6 (±13.9) | 26.7 (±3.3) | 24.4 (±8.4) | 26.7 (±12.0) | 30.0 (±3.3) |
| Window Close | 3.3 (±3.3) | 10.0 (±0.0) | 25.6 (±36.0) | 14.4 (±13.5) | 16.7 (±15.3) | 20.0 (±15.3) | 21.1 (±13.5) | 20.0 (±12.0) | 34.4 (±13.5) | 25.6 (±8.4) |
| Window Open | 25.6 (±5.1) | 44.4 (±18.4) | 56.7 (±11.5) | 74.4 (±1.9) | 71.1 (±13.5) | 74.4 (±7.7) | 81.1 (±11.7) | 73.3 (±8.8) | 72.2 (±15.0) | 74.4 (±3.8) |
| AVG | 14.7 (±0.6) | 27.7 (±1.9) | 33.5 (±2.2) | 43.5 (±2.6) | 44.2 (±4.5) | 51.1 (±4.0) | 51.5 (±3.0) | 52.0 (±5.0) | 52.9 (±5.2) | 53.1 (±3.8) |

Table A8: **MetaWorld Task Performance.** We provide a detailed list of task performance for the leftmost plot in Figure 4. We report the mean success rate and standard deviation aggregated over 3 seeds each, across 10 iterations. We reiterate that none of these 12 tasks had been seen a priori during initial in-domain video model training.

## D  GENERALIZATION CAPABILITIES BETWEEN VISUAL PLANNING AND ACTION-PREDICTIVE BC

Our results suggest that visual planning has some default generalization and self-improvement benefits for decision-making compared to using direct action-prediction BC policies. We hypothesize

that modeling consistent visual motions can extract more training signal from the provided data than modeling action sequences directly, as there is more supervisory training signal from pixels than low-dimensional actions. Indeed, as shown in the MetaWorld portion of Figure 2, for initial iteration 0 on a novel task concerning an unseen object, the visual planner still manages to generate coherent motions for the robot arm even if it blurs out the specific novel object interaction. These coherent motions, even if the specific object interaction is not modeled correctly (or even blurred) initially, can still be accurately translated by the IDM into meaningful robot actions; thus visual planning may have additional generalization benefits. On the other hand, the basic BC policy does not model visual details but predicts a sequence of actions entirely from the conditioning frame; having overfit to its training set, when faced with a novel object in the scene it may predict highly suboptimal actions, thus leading to poorer generalization.

## E  FULL METAWORLD SUBOPTIMAL RESULTS

As mentioned in Section 4.6, we evaluate SILVR on 12 unseen MetaWorld tasks with suboptimal initial in-domain data. We provide a detailed task performance breakdown in Figure A1 and Table A9. We observe that most tasks, as well as average performance, exhibit an improving trend overall across iterations, demonstrating the robustness of SILVR to initial data quality.

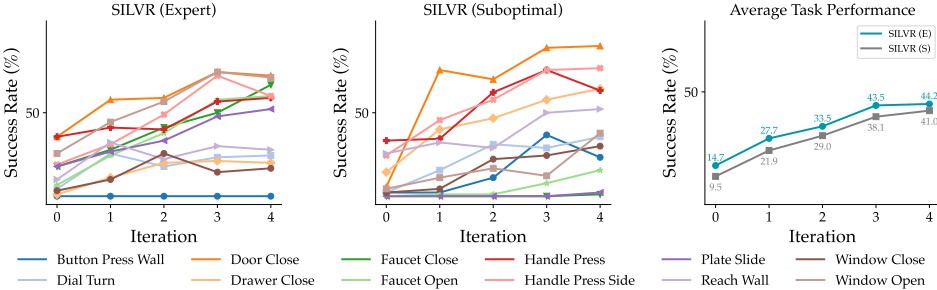

Figure A1: **SILVR Curves with suboptimal in-domain data.** For each task, we plot the mean success rate across 5 iterations, aggregated over 3 seeds.

| Iteration | 0 | 1 | 2 | 3 | 4 |
|---|---|---|---|---|---|
| Button Press Wall | 2.2 ($\pm$1.9) | 2.2 ($\pm$3.8) | 11.1 ($\pm$10.2) | 36.7 ($\pm$32.1) | 23.3 ($\pm$20.8) |
| Dial Turn | 2.2 ($\pm$1.9) | 15.6 ($\pm$12.6) | 31.1 ($\pm$13.5) | 28.9 ($\pm$23.4) | 35.6 ($\pm$11.7) |
| Door Close | 5.6 ($\pm$3.8) | 75.6 ($\pm$18.4) | 70.0 ($\pm$8.8) | 88.9 ($\pm$7.7) | 90.0 ($\pm$3.3) |
| Drawer Close | 14.4 ($\pm$6.9) | 40.0 ($\pm$21.9) | 46.7 ($\pm$20.3) | 57.8 ($\pm$29.1) | 64.4 ($\pm$28.3) |
| Faucet Close | 0.0 ($\pm$0.0) | 0.0 ($\pm$0.0) | 0.0 ($\pm$0.0) | 0.0 ($\pm$0.0) | 1.1 ($\pm$1.9) |
| Faucet Open | 0.0 ($\pm$0.0) | 1.1 ($\pm$1.9) | 1.1 ($\pm$1.9) | 7.8 ($\pm$8.4) | 15.6 ($\pm$16.8) |
| Handle Press | 33.3 ($\pm$5.8) | 34.4 ($\pm$6.9) | 62.2 ($\pm$15.0) | 75.6 ($\pm$13.9) | 63.3 ($\pm$21.9) |
| Handle Press Side | 24.4 ($\pm$1.9) | 45.6 ($\pm$9.6) | 57.8 ($\pm$10.2) | 75.6 ($\pm$20.4) | 76.7 ($\pm$5.8) |
| Plate Slide | 0.0 ($\pm$0.0) | 0.0 ($\pm$0.0) | 0.0 ($\pm$0.0) | 0.0 ($\pm$0.0) | 2.2 ($\pm$1.9) |
| Reach Wall | 25.6 ($\pm$5.1) | 32.2 ($\pm$6.9) | 28.9 ($\pm$7.7) | 50.0 ($\pm$6.7) | 52.2 ($\pm$6.9) |
| Window Close | 2.2 ($\pm$1.9) | 4.4 ($\pm$1.9) | 22.2 ($\pm$12.6) | 24.4 ($\pm$21.7) | 30.0 ($\pm$23.3) |
| Window Open | 4.4 ($\pm$1.9) | 11.1 ($\pm$6.9) | 16.7 ($\pm$12.0) | 12.2 ($\pm$6.9) | 37.8 ($\pm$22.7) |
| AVG | 9.5 ($\pm$0.3) | 21.9 ($\pm$0.9) | 29.0 ($\pm$2.3) | 38.1 ($\pm$4.9) | 41.0 ($\pm$4.4) |

Table A9: **SILVR Performance with Suboptimal Initial Data.** For each task, we report the mean success rate and standard deviation aggregated over 3 seeds, across 5 iterations.

Additionally, we provide BCIL and DSRL performance with suboptimal in-domain data in Table A10. Surprisingly, we observe that BCIL benefits more from suboptimal initialization. We hypothesize that the exploration brought by initial random actions is crucial for bootstrapping the Diffusion Policy performance in BCIL.

| Iteration | 0 | 1 | 2 | 3 | 4 |
|---|---|---|---|---|---|
| DSRL | 7.4 (±3.8) | 7.1 (±0.6) | 8.6 (±1.3) | 7.8 (±1.0) | 8.1 (±0.8) |
| BCIL | 8.1 (±1.0) | 21.6 (±0.9) | 29.1 (±2.3) | 37.2 (±6.1) | 39.6 (±5.3) |

Table A10: **Baseline Performance with Suboptimal Initial Data.** We report the mean success rate and standard deviation across 12 unseen tasks, aggregated over 3 seeds each.

## F METAWORLD COMPONENT ABLATIONS

| Iteration | 0 | 1 | 2 | 3 | 4 |
|---|---|---|---|---|---|
| Finetuning All (SILVR) | 14.7 (±0.6) | 27.7 (±1.9) | 33.5 (±2.2) | 43.5 (±2.6) | 44.2 (±4.5) |
| No IDM Finetuning | 14.1 (±1.6) | 22.2 (±3.1) | 24.6 (±2.5) | 24.8 (±1.7) | 26.8 (±1.9) |
| Only IDM Finetuning | 15.0 (±1.8) | 24.4 (±3.2) | 27.7 (±3.5) | 26.9 (±4.1) | 29.8 (±3.4) |

Table A11: **Component Update Ablations.** We report the mean success rate and standard deviation across 12 unseen tasks, aggregated over 3 seeds each.

We investigate the importance of finetuning different visual planning components with respect to online experience on final task performance in the MetaWorld suite. We find that in contrast with the real-world experiments, keeping the IDM frozen for the MetaWorld suite struggles to adjust to novel objects and motions, resulting in modest performance gains even when the video model improves. Similarly, we have found that when the IDM is finetuned but the video model is not, there are indeed some performance gains most likely from better translation of the base generalization ability of the video model; but performance also saturates quickly. Thus, under situations with heavy novelty such as across objects and motions, updating both components with respect to online experience is critical. Meanwhile, in real-world experiments, we observed that the initially trained IDMs are inherently robust to novel tasks, in which many motions learned from seen tasks can be reused, and can be utilized directly to great effect without additional finetuning.

# G  ADDITIONAL PLAN VISUALIZATIONS

We show additional visual plans for SILVR, across multiple environments and tasks, along with their execution results.

## G.1  SILVR WITH GROUND-TRUTH FILTERING

Visual plans and their executions for SILVR with Ground-Truth filtering are illustrated below.

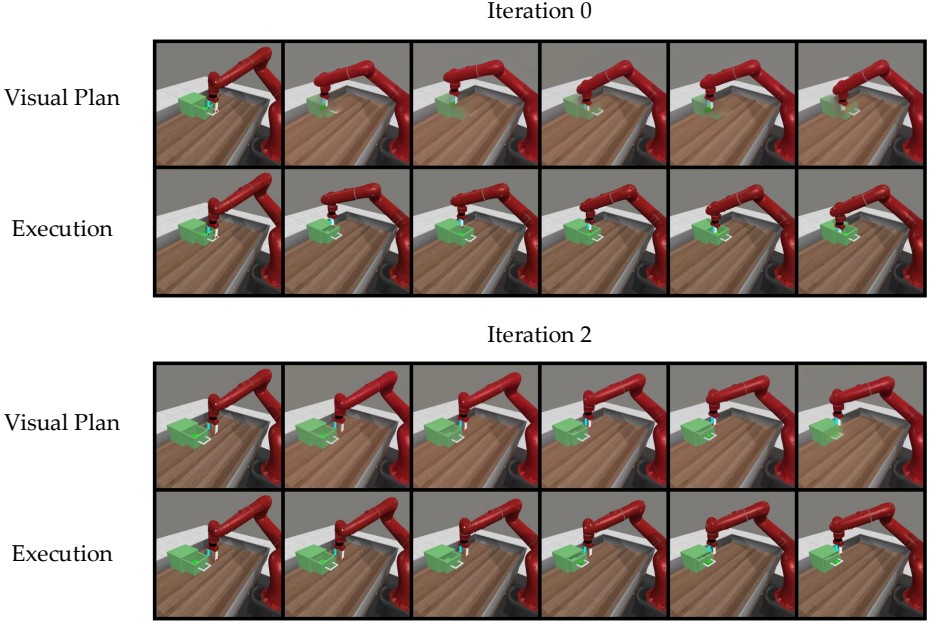

Figure A2: **SILVR on Drawer Close with Ground-Truth filtering.**

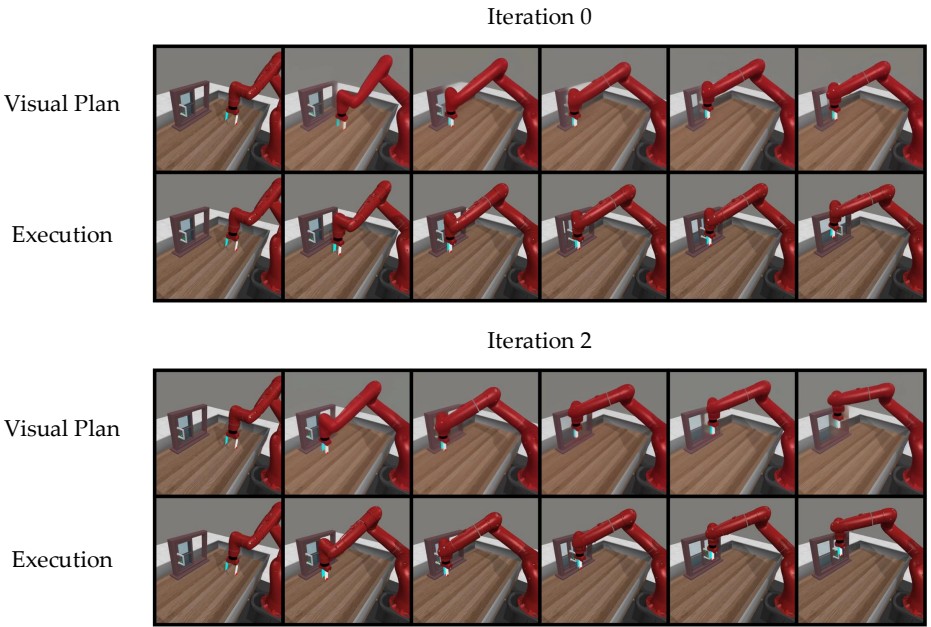

Figure A3: **SILVR on Window Close with Ground-Truth filtering.**

Iteration 0

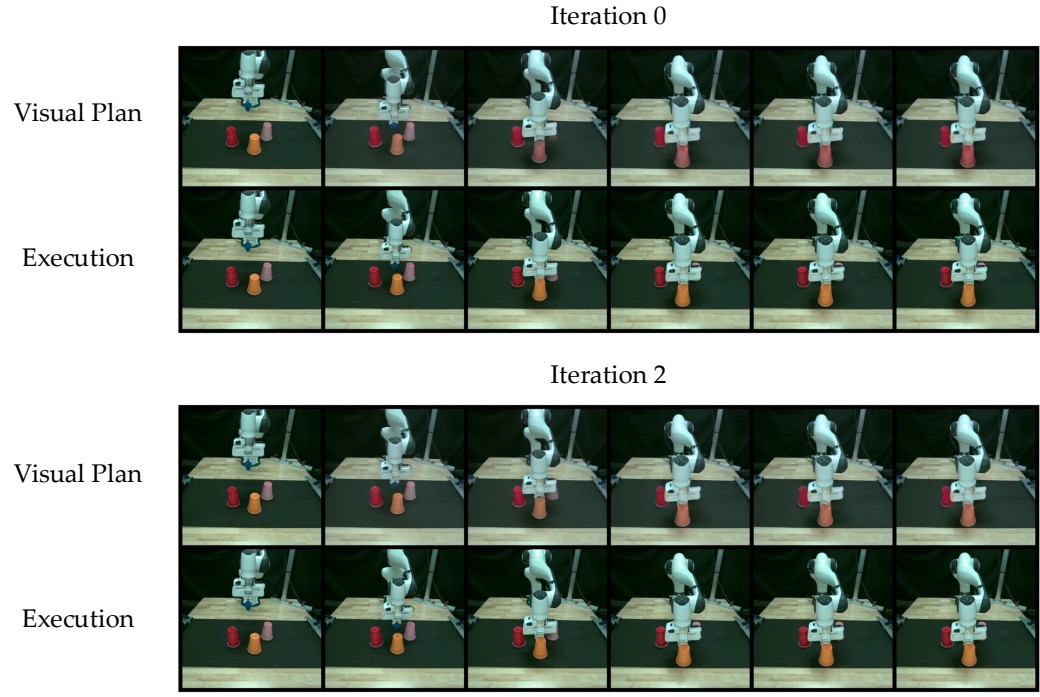

Figure A4: **SILVR on Orange Cup Pushing (Red/Pink/Orange) with Ground-Truth filtering.**

Iteration 0

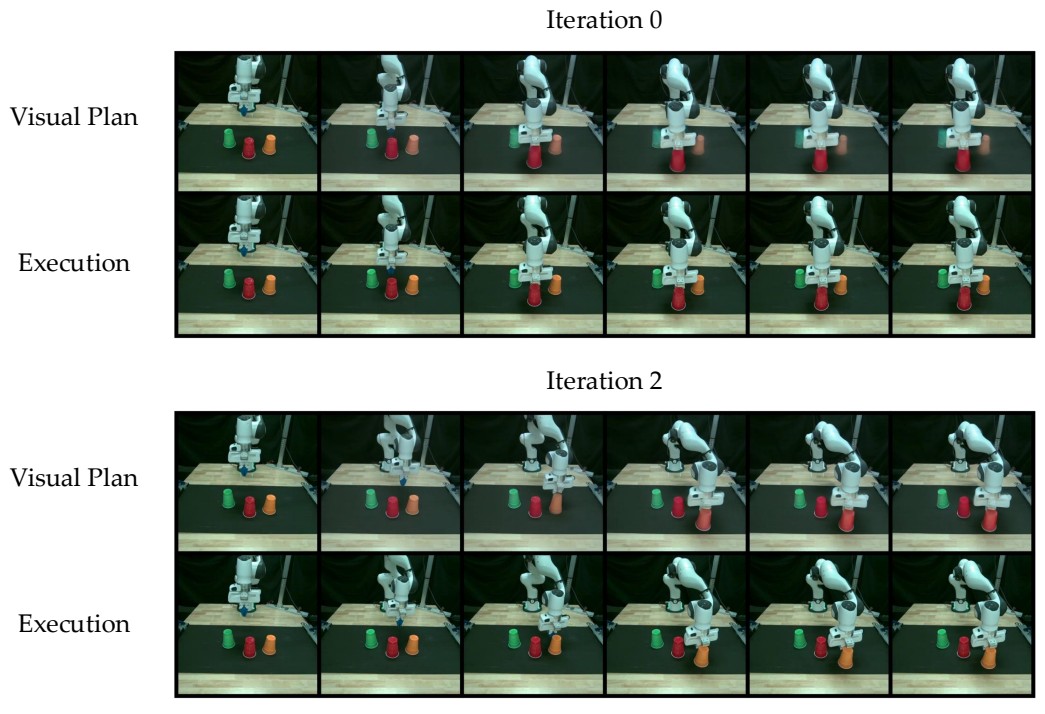

Figure A5: **SILVR on Orange Cup Pushing (Red/Green/Orange) with Ground-Truth filtering.**

Iteration 0

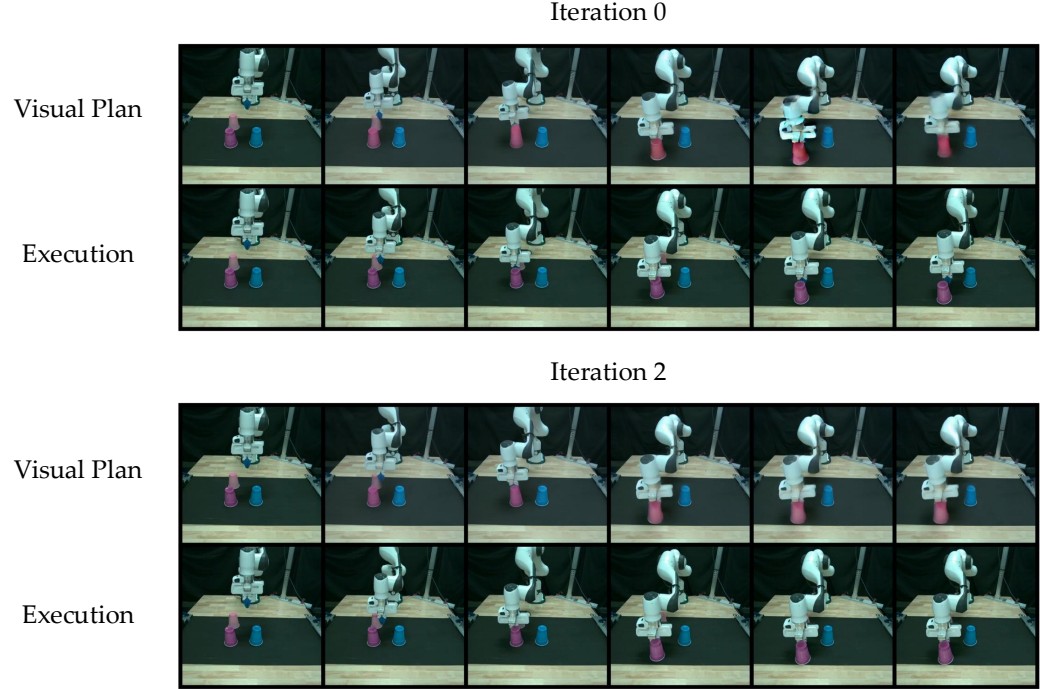

Figure A6: **SILVR on Purple Cup Pushing (Blue/Pink/Purple) with Ground-Truth filtering.**

Iteration 0

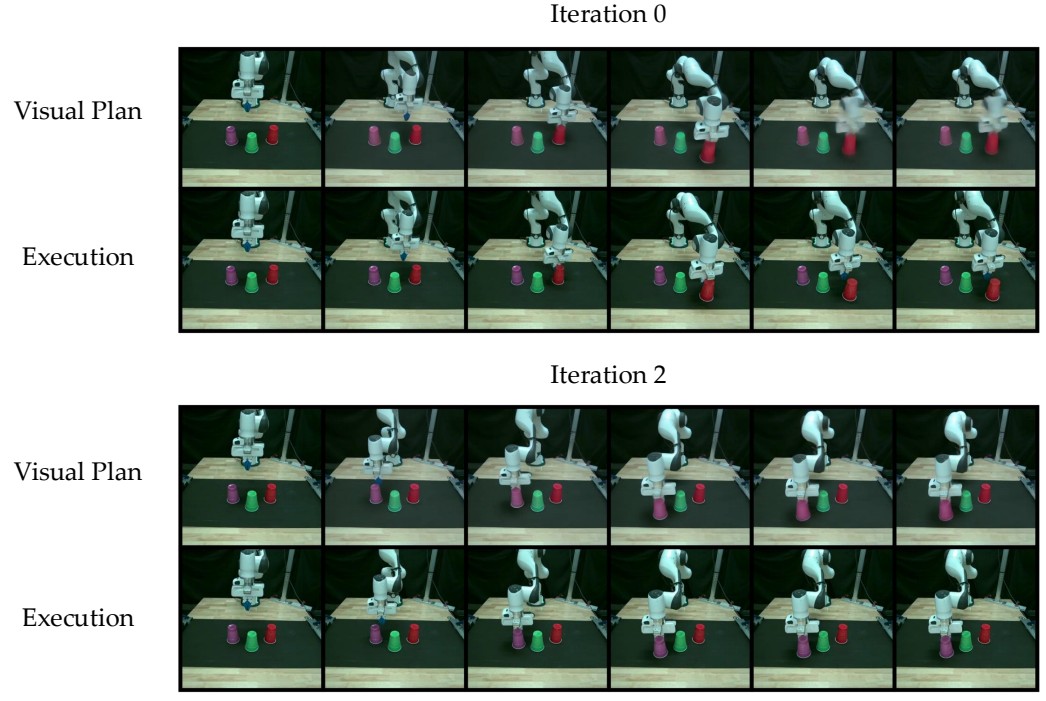

Figure A7: **SILVR on Purple Cup Pushing (Red/Green/Purple) with Ground-Truth filtering.**

Iteration 0

Visual Plan

Execution

Iteration 2

Visual Plan

Execution

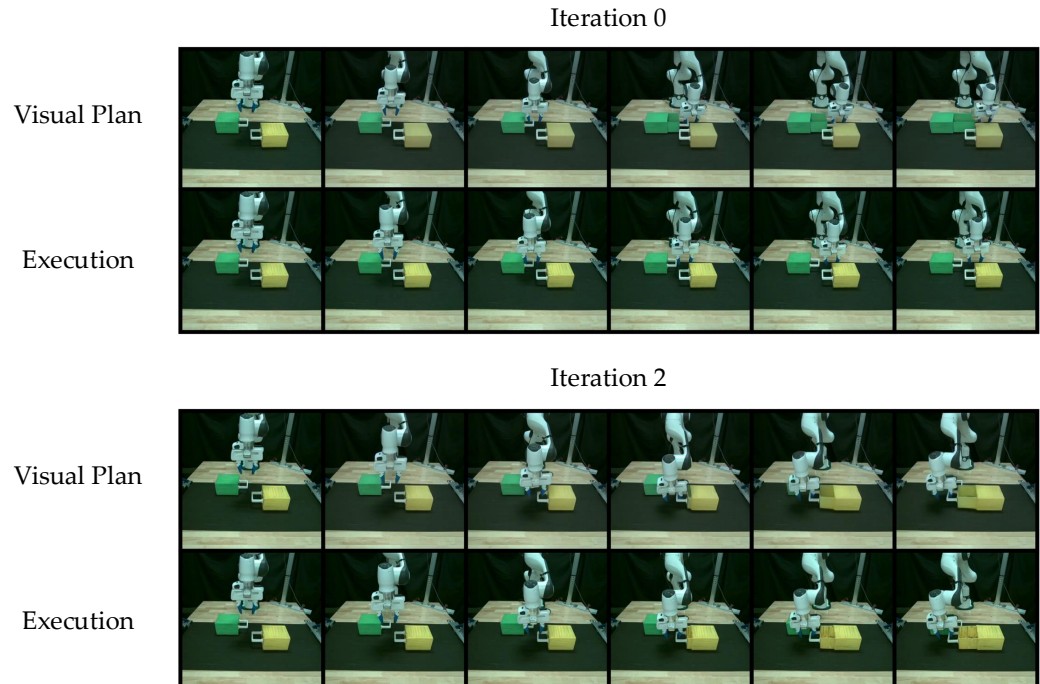

Figure A8: **SILVR on Yellow Drawer Opening (Yellow/Green) with Ground-Truth filtering.**

Iteration 0

Visual Plan

Execution

Iteration 2

Visual Plan

Execution

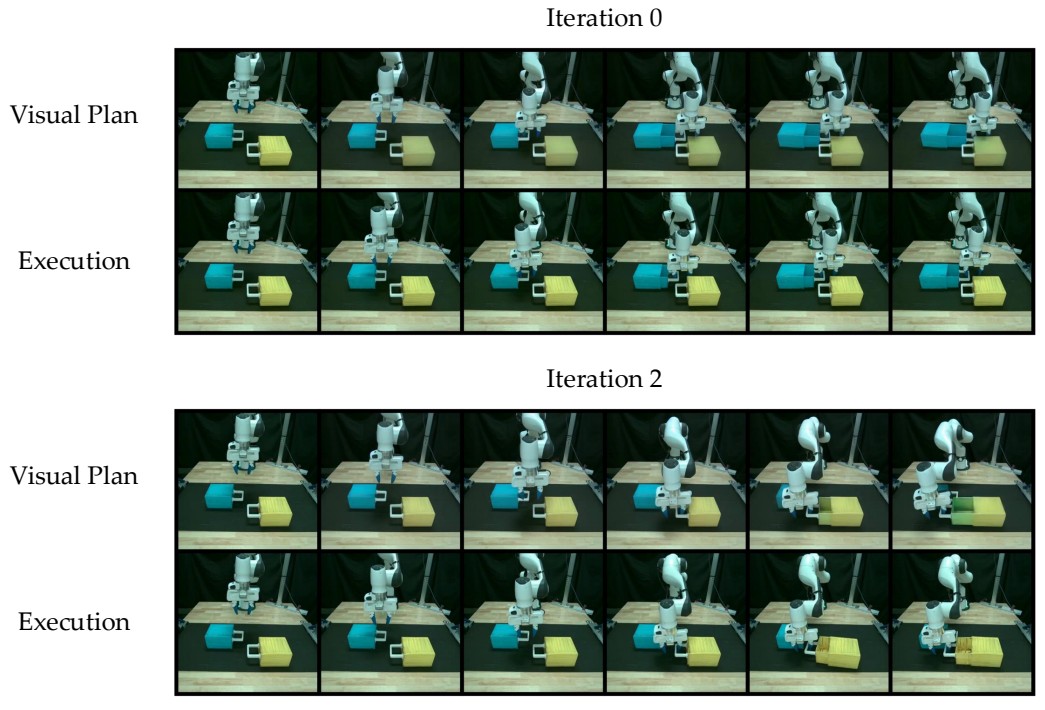

Figure A9: **SILVR on Yellow Drawer Opening (Yellow/Blue) with Ground-Truth filtering.**

## G.2 SILVR WITHOUT FILTERING

Visual plans and their executions for SILVR without filtering are illustrated below.

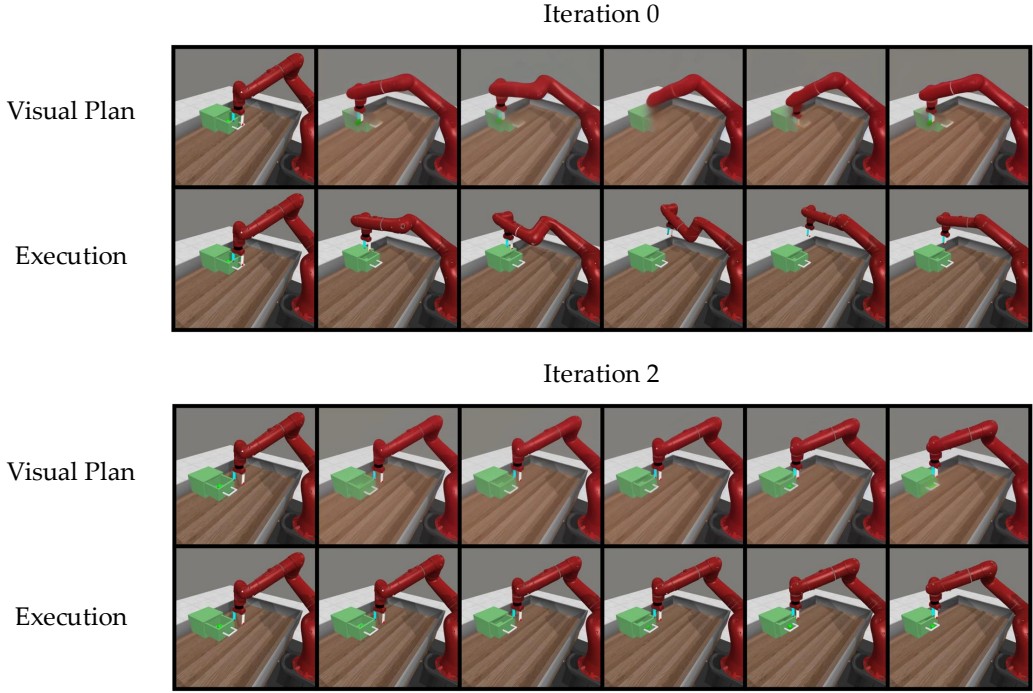

Figure A10: **SILVR on Drawer Close without filtering.**

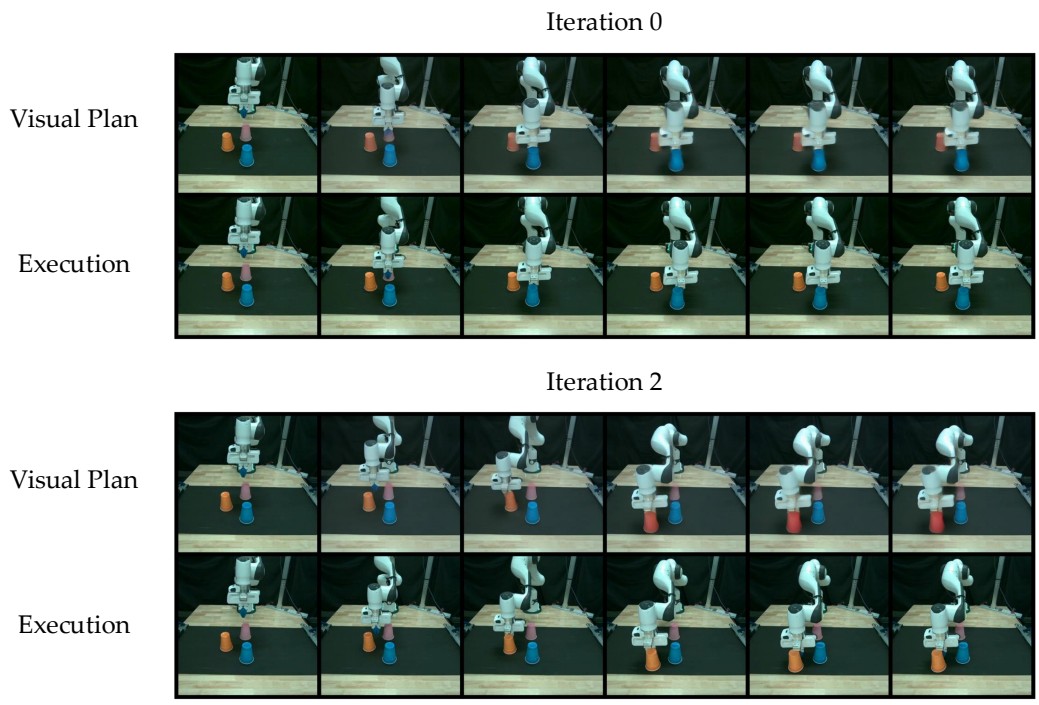

Figure A11: **SILVR on Orange Cup Pushing (Blue/Pink/Orange) without filtering.**

## G.3   Corrective Influence of Internet-Scale Video Model Adaptation

In-Domain Only

Visual Plan

Execution

Adapted w/ AnimateDiff

Visual Plan

Execution

Figure A12: **Push Orange Cup**

In-Domain Only

Visual Plan

Execution

Adapted w/ AnimateDiff

Visual Plan

Execution

Figure A13: **Push Purple Cup**

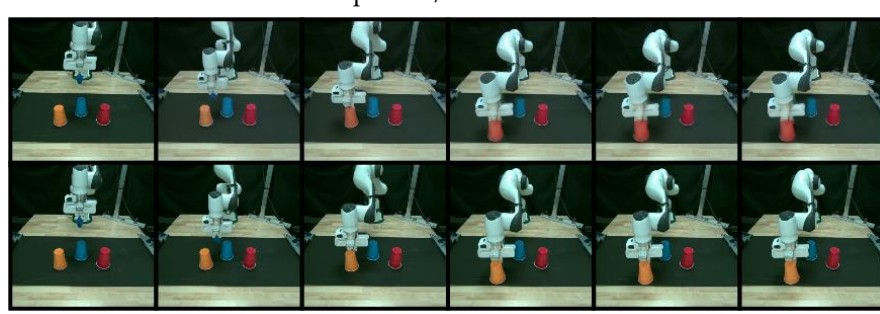

In-Domain Only

Visual Plan

Execution

Adapted w/ AnimateDiff

Visual Plan

Execution

Figure A14: **Open Yellow Drawer**

In-Domain Only

Visual Plan

Execution

Adapted w/ AnimateDiff

Visual Plan

Execution

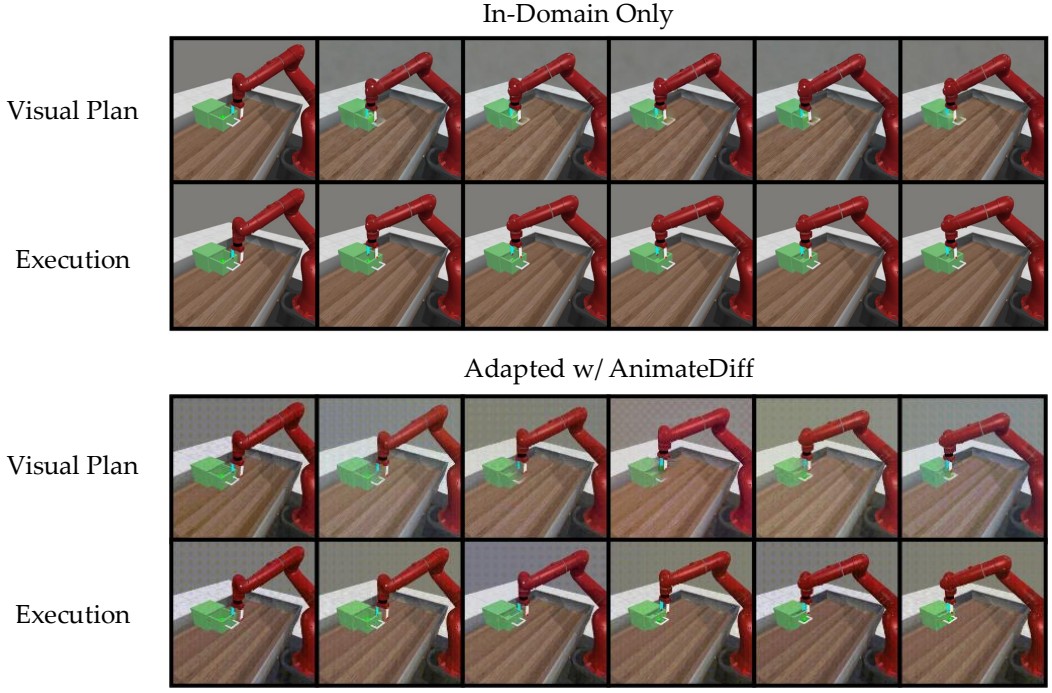

Figure A15: **Drawer Close**

### G.4 Long-Horizon Task Evaluation

Whereas the experiments thus far have demonstrated how SILVR can iteratively improve performance on single tasks, a natural question to consider is how applicable it is to complex tasks. We note that many long-horizon tasks can be broken down into atomic tasks with shorter horizons, for which SILVR can already demonstrate clear and robust self-improvement behaviors. We thus investigate how SILVR, by learning individual abilities, naturally facilitates the direct iterative improvement of a sequential long-horizon tasks where success rate depends on the full completion of behaviors in a specific order.

In particular, we construct an experiment of pushing cups in an order specified by natural language. For example, in Figure A16 the prompt is "Push Cups in the order of Red, Orange, Green", where success is only satisfied if done so in the correct order. We evaluate SILVR checkpoints automatically using a VLM, implemented as Gemini 2.5, to determine what atomic skill (e.g., pushing the cup of a specific color) to currently execute from the instruction sequence and when to switch to the subsequent one by querying if the visual execution of the current atomic skill was successful.

We find that over successive iterations, despite only applying SILVR to individual atomic tasks, the performance on the overall long-horizon tasks improves. This suggests that for arbitrary complex long-horizon tasks of interest, SILVR can be combined with an efficient subtask segmentation mechanism to improve individual performance on atomic behaviors and thus improve the final overall performance on long-horizon sequential compositions.

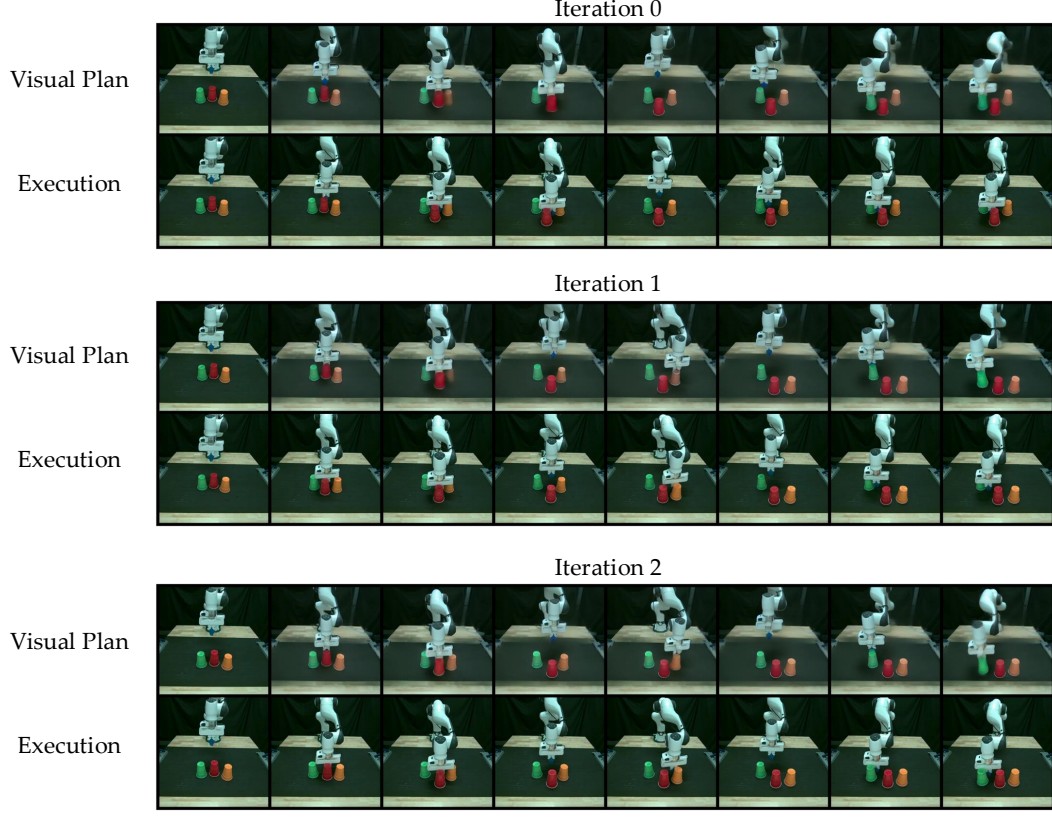

Figure A16: **Push Cups in the order of Red, Orange, and Green.** The final-iteration visual planner completed the long-horizon task, whereas those from the first two iterations failed.

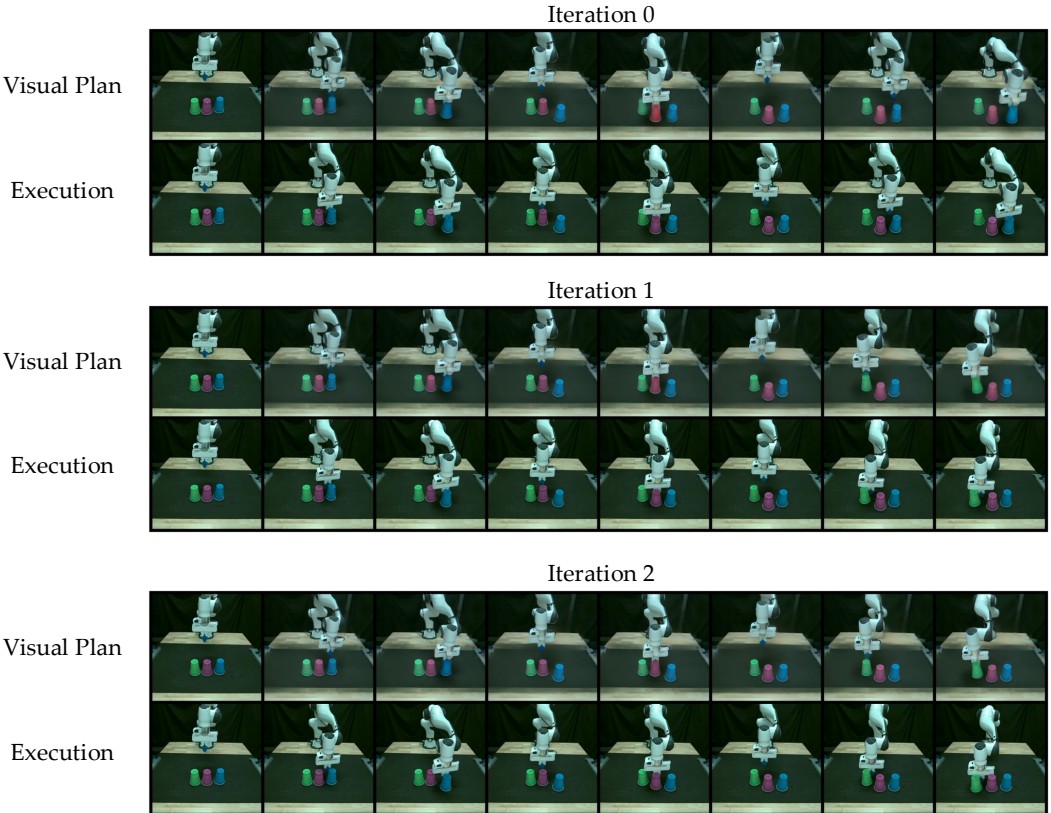

Figure A17: **Push Cups in the order of Blue, Purple, and Green.** The visual planners of the last two iterations completed the long-horizon task, whereas the one from the first iteration failed.

### G.5 VISUALIZING FAILURE CASES OF FINAL-ITERATION VISUAL PLANNERS

Below we visualize the failure cases from the final-iteration visual planners across both real-world and simulation task settings. Most real-world failures arise from semantically incorrect visual plans, in which the robot arm attempts to push the cup (Figure A18 and A19) or open the drawer (Figure A20) of a wrong color, whereas in simulation, we observe a mixture of execution (Figure A21) and semantic (Figure A22) errors, both of which can contribute to unsuccessful task completion.

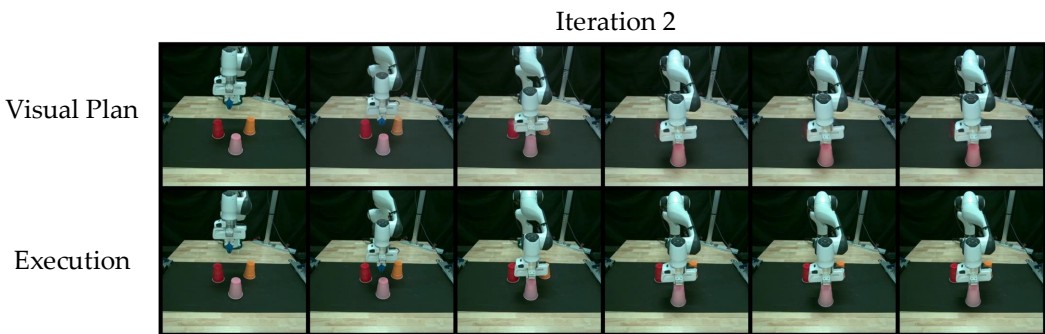

Figure A18: **Push the Orange Cup (Iteration 2)**

Iteration 2

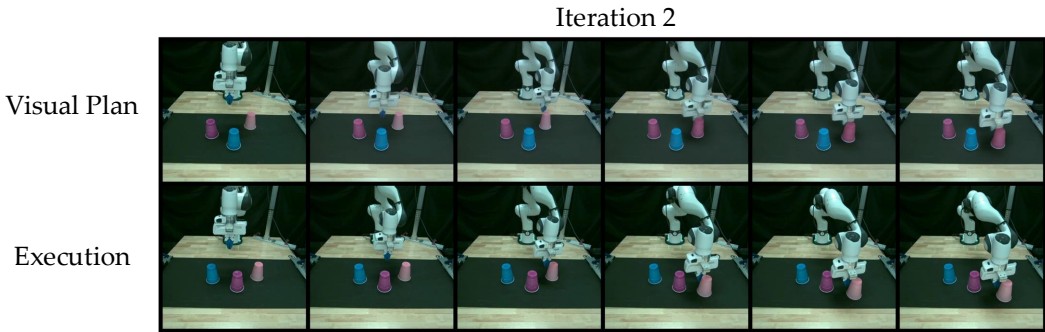

Figure A19: **Push the Purple Cup (Iteration 2)**

Iteration 2

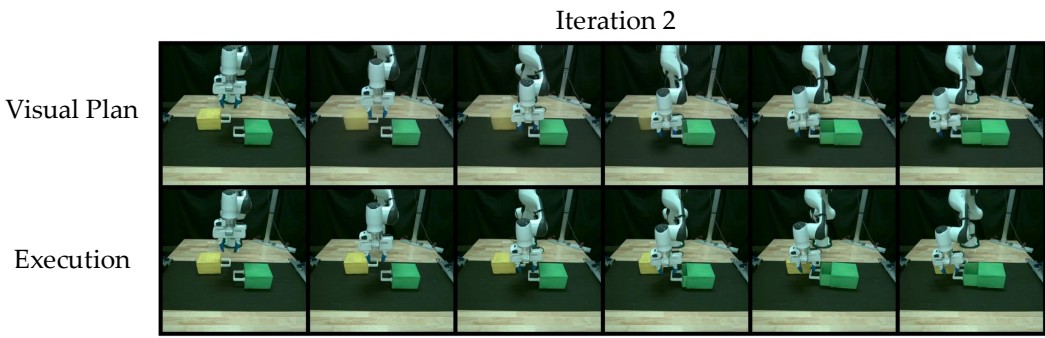

Figure A20: **Open the Yellow Drawer (Iteration 2)**

Iteration 9

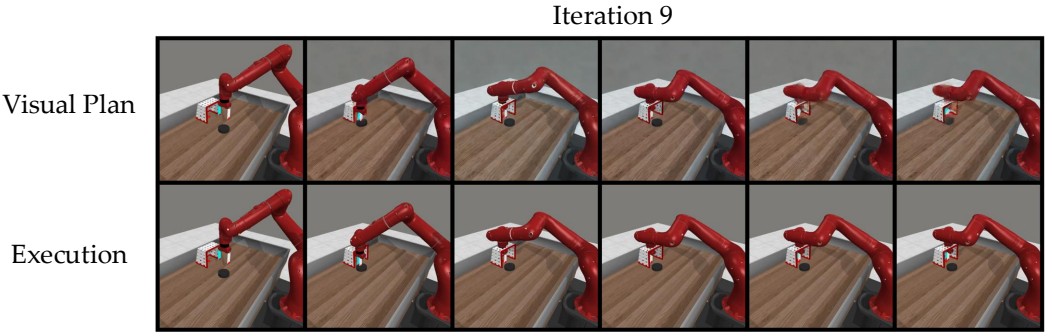

Figure A21: **Plate Slide (Iteration 9)**

Iteration 9

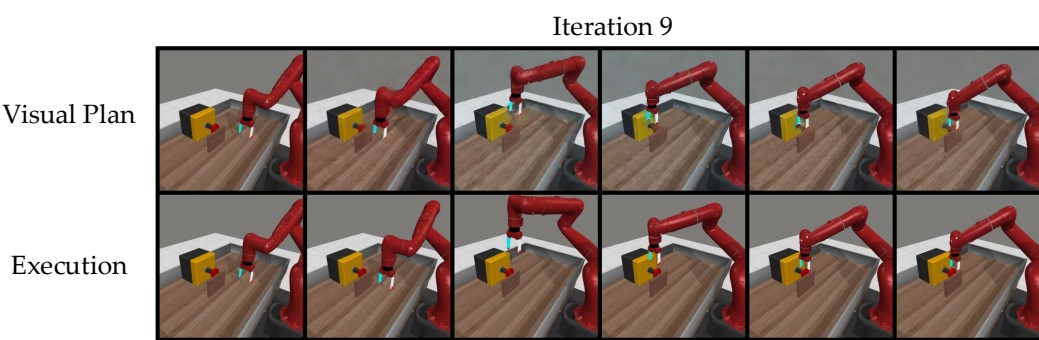

Figure A22: **Button Press Wall (Iteration 9)**

