# OpenReview forum: "Self-Improving Loops for Visual Robotic Planning"
_ICLR.cc/2026/Conference — ICLR 2026 Poster_

### Official Review · Reviewer_2K1s · 2025-10-22

**Soundness:** 3
**Presentation:** 1
**Contribution:** 2
**Rating:** 4
**Confidence:** 4

**Summary:**

The paper proposes **SILVR (Self-Improving Loop for Visual Robotic Planning)**, a framework that treats a video generative model as a visual planner and **iteratively improves it with self-collected experience**. Concretely, an in-domain video model produces future-frame plans given a text prompt; a separately trained **inverse dynamics model (IDM)** converts consecutive planned frames into executable actions that are run in the environment, after which successful rollouts are filtered and used to **finetune both the in-domain video model and the IDM**—then the loop repeats.

**Strengths:**

1. The authors propose a simulator-driven self-optimization approach for video generation, enabling the success rate on corresponding tasks to increase with additional iterations.
2. They evaluate SILVR’s generalization and robustness across multiple environments. In simpler settings, SILVR can accomplish the target tasks to a certain extent and shows a degree of robustness and generalization.

**Weaknesses:**

1. Requires an initially solvable task; limited gains when the starting success rate is very low.
2. Real-world gains depend on an internet video prior and are modest/unstable without it, with few reported iterations.
3. Evaluation focuses on short-horizon/simple tasks, leaving feasibility on harder or long-horizon tasks unclear.
4. Minor presentation issue: Some figures are hard to read in the PDF due to lack of detail explaination for some signals.

**Questions:**

1. Why do some tasks trained from suboptimal initial data outperform the expert-initialized setting?
2. Why does the SILVR-distilled Diffusion Policy (DP) achieve a higher success rate than the final SILVR visual planner?

---

> ### Author Response · Authors · 2025-11-22
>
> We thank Reviewer 2K1s for their comments and helpful feedback on our work. Below, we seek to both address the reviewer’s listed weaknesses and answer the posed questions:
>
> **On initial solvability:** We agree with the reviewer that as vanilla SILVR depends on filtering, initial solvability is needed.  Fortunately, SILVR can naturally utilize large-scale pretrained text-to-video models, which contribute powerful zero-shot text-alignment signals. We hypothesize that web-scale video data, demonstrating diverse task completions in different environments and embodiments, will essentially resolve the initial solvability problem when properly adapted to the target environments.
>
> Secondly, the “cold start” problem is a known general problem for decision-making (e.g. reinforcement learning).  We believe that another way to mitigate it is through improving exploration capabilities under the visual planning framework; through high-quality exploration techniques, the chance of collecting successful experience can be improved, at which point SILVR can be subsequently applied.  Whereas the scope of SILVR focuses on self-improvement for visual planning techniques, we believe exploration is a worthwhile direction to investigate for future work.
>
> Lastly, “initial solvability” is relative; even if it does not solve the task according to an oracle, self-improvement can still be gained when using a noisy filtering strategy - for example, if a rollout was deemed “initially solvable” as judged by an imperfect oracle such as Gemini.  Even if such rollouts do not actually solve the task, the noisy VLM oracle may still identify task-relevant behaviors that are worth observing as online experience.  In fact, we even show that a self-improvement trend occurs in both MetaWorld and the real world without filtering (Figure 5, Section 4.4).  SILVR thus has the potential to still learn meaningful behaviors even with limited initial solvability; even suboptimal trajectories can supply useful information to improve the in-domain model (and thus the entire composed visual planner) to produce more performant plans over subsequent iterations.
>
> **On internet video prior for real-world gains:** Modeling the visual real world is difficult, and as the reviewer noted in the previous point, initial solvability can be leveraged quickly for self-improvement through SILVR.  In this light, we highlight how SILVR can directly integrate large-scale pretrained video models as a benefit, as it can help the initial solvability problem in the real world by utilizing powerful priors over text-alignment and motions.
>
> **On long-horizon tasks:** We focus on demonstrating generalization and self-improvement across novel tasks, rather than solving long-form tasks.  For example in the MetaWorld suite, despite being individually simple, accomplishing such tasks in a zero-shot manner and learning effectively through few *self-collected* demonstrations is difficult, as such novel tasks often involve previously unseen objects or different motions in order to solve.  We agree however that solving long-horizon tasks is interesting, and as SILVR is based on visual planning, we hypothesize that its performance on such tasks is fundamentally tied to the capabilities of video models to perform instruction following accurately and over long contexts.
>
> At the same time, many long-horizon tasks can be essentially tackled by dividing it into atomic tasks with shorter horizons on which our framework, as the reviewer noted, already demonstrates robust self-improvement behaviors.  SILVR can therefore potentially be utilized with an efficient instruction/skill/temporal segmentation mechanism, to improve individual performance on atomic tasks and thus improve the final combined overall performance on long-horizon tasks.
>
> **On signals in the figures:**  We update the draft with additional caption clarifications and annotations for a variety of included figures to aid in understanding.  For figures with numerical results we include the numbers on top of the bullet points for hopefully added clarity and readability; their values correspond to the labels on the y-axis (predominantly success rate).  If there are any Figures in particular that the reviewer believes would benefit from additional clarification we would be happy to iterate on them.

---

> ### Author Response · Authors · 2025-11-22
>
> **On suboptimal initial data performance:** In Figure 6 we highlight the top three novel tasks that benefit from expert demonstrations (Faucet Open, Faucet Close, Plate Slide) in comparison to suboptimal demonstrations; we also highlight the top three novel tasks that benefit more from initializing from suboptimal demonstrations rather than expert ones (Drawer Close, Reach Wall, Button Press Wall).  Suboptimal demonstrations were collected with a large percentage of random actions; and as such we expect a model trained on such data to output more random actions and behave in a more exploratory manner.  As described in Section 4.5, we thus hypothesize that such tasks benefit more from random exploration (as in, the robot arm can move haphazardly and close the drawer or find the button), whereas learned complex motions from expert data may be confidently incorrect for solving such tasks.  On the other hand, more complex tasks such as sliding a plate into a goal or turning a faucet may be difficult to achieve from initially suboptimal motion demonstrations and active exploration, but can potentially benefit from coherent complex motions from other tasks.  This balance between exploration and reuse of learned skills can potentially be addressed by *principled* exploration techniques for visual planning, which we identify as promising future work to investigate.
>
> **On distilled policy performance:** We agree that this is an interesting finding, particularly as we ensured the amount of demonstrations utilized in distillation is kept consistent with those used in BCIL and SILVR for fair comparison across final performance (distillation details described in Appendix B).  Similar findings have recently been discovered in LLM literature, where distilling a language model into another comparably sized language model has been found to be better than training on the original dataset (Section 6 of [1]).  We hypothesize that in this visual planning policy distillation, a similar phenomenon is occurring as well.  The improved performance of the distilled policy is a surprising result, but one that we consistently observed (Iteration 4 in Table 1 as well as Iteration 9 in Figure 4).
>
>
> [1] Kim et al., Pre-training under infinite compute, 2025.

---

> > ### Comment · Reviewer_2K1s · 2025-11-27
> >
> > Thank you very much for the detailed responses. However, I believe that tasks such as Drawer Close, Reach Wall, and Button Press Wall are not novel. Therefore, I will maintain my original score.

---

> > > ### Author Response · Authors · 2025-11-27
> > >
> > > We would like to thank the reviewer for their timely reply! As noted, the three tasks may seemingly share certain similarities with some of the seen tasks. However, we believe that these tasks do meaningfully contribute to the novelty of our evaluation set, and enable us to measure task generalization from different aspects:
> > >
> > > (1) *Drawer Close* requires a completely different motion from its seen drawer variant to achieve success.
> > >
> > > (2) *Reach Wall* forces the visual planner to adapt to an environment with a previously unseen visual distractor (a large wall unencountered across any seen tasks) and generate coherent, obstacle-aware motions.
> > >
> > > (3) *Button Press Wall* features entirely novel objects and motions, as the red button was not observed in any of the seen tasks (the closest task in the seen suite is Coffee Push, which features an entirely different button shape, color, appearance, and also has no wall to navigate around).
> > >
> > > Moreover, these tasks indeed pose genuine challenges for task generalization as indicated by their performance on Iteration 0. As such, we believe these tasks satisfy a valid and meaningful definition of task novelty.

---

### Official Review · Reviewer_UiiZ · 2025-10-27

**Soundness:** 3
**Presentation:** 3
**Contribution:** 3
**Rating:** 6
**Confidence:** 3

**Summary:**

In this work, the authors propose a method named Self-Improving Loops for Visual Robotic Planning (SILVR), which iteratively updates in-domain video model on self-produced trajectories to improve its performance for a specified task of interest. The authors thoroughly demonstrate how SILVR establishes connection between in-domain video model and high-performing visual planner for specified novel robotic control task. Extensive evaluations on the MetaWorld task suite and real-world robot arm verify the effectiveness of SILVR.

**Strengths:**

1. The proposed SILVR is straightforward and easy to follow, leveraging iterative fine-tuning of a visual planner on self-collected experience without complex reward engineering.
2. The paper provides extensive experiments across both simulated (MetaWorld) and real-world (Franka Panda arm) settings, demonstrating broad applicability and robustness.
3. SILVR shows significant performance improvement on unseen tasks (up to 285% in MetaWorld) and outperforms reinforcement learning and behavior cloning baselines in sample efficiency.
4. The method is robust to suboptimal initial data, does not require ground-truth rewards (works with VLM-based filtering), and can effectively incorporate internet-scale video priors for real-world generalization.

**Weaknesses:**

Since that I am not an expert in this field, here are a few potential weaknesses I observed from the paper:
1. While the method is well-executed and effective, the core idea of a self-improving loop—fine-tuning a model on its own successful outputs—is a well-established concept in other areas like large language models. Applying this established concept to visual planning, while practical, may be perceived as a solid incremental advancement rather than a foundational shift in methodology.
2.  The real-robot experiments (pushing a colored cup, opening a colored drawer), while important for demonstrating real-world applicability, are relatively simple table-top manipulations. It remains unclear how well SILVR would scale to tasks with more complex dynamics, longer time horizons, or tasks requiring intricate multi-step reasoning and tool use.
3. The self-improving loop implicitly assumes the initial model can achieve a non-trivial success rate to collect useful data. For tasks that are too distinct from the initial training distribution, the model might fail completely at iteration zero, preventing any improvement and creating a "cold-start" problem that SILVR fails to address.

**Questions:**

See weaknesses

---

> ### Author Response · Authors · 2025-11-22
>
> We thank Reviewer UiiZ for their detailed comments and thorough questions. We seek to address their concerns below:
>
> **On self-improvement:** We agree that finetuning a model on its self-collected experience is an area of study with a long history; indeed, reinforcement learning seeks to accomplish this exact aim for policies.  Visual planning is a separate paradigm where actions are not generated from a policy directly, but through a video model and an inverse dynamics model; and although the aim of learning from online experience is a general desire (across environments, even language models), we believe our novelty lies in proposing a framework for enabling such capabilities for visual planning.  In particular, we identify that the separation of an in-domain model and large-scale pretrained model not only achieves high zero-shot generalization but also facilitates self-improvement over novel tasks; we also show how self-improvement for visual planning can outperform reinforcement learning based approaches like DSRL.  Furthermore, we utilize our framework to provide novel insights into self-improvement for visual planning, such as reliance and robustness on filtering strategies.
>
> Lastly, as [discussed with Reviewer dWn2](https://openreview.net/forum?id=SzUgx5r3wy&noteId=dC8uDMN5aa), only using a large-scale video model such as AnimateDiff as the planner in the SILVR framework (where successfully collected online experience is reused to refine the video model) fails on real-world tasks, and thus fails to demonstrate self-improvement (visualizations provided on the [updated website](https://silvr-anonymous.github.io/), in section “Visual Planning with AnimateDiff Only”).  This highlights that SILVR’s utilization of score composition with an in-domain model is critical for achieving initial performance and facilitating subsequent self-improvement throughout iterations of the loop, a novel and crucial finding for understanding how visual planners can achieve self-improvement for novel tasks of interest.
>
> **On tasks with increased complexity:** SILVR studies how online experience can be used to improve visual planning techniques; but we agree with the reviewer that how visual planning as an approach itself can model complex dynamics, multi-step reasoning, etc. is an interesting question.  Indeed, complex dynamics, multi-step reasoning, long horizon with sparse rewards, etc. are all known as challenges in traditional reinforcement learning as well.  We hypothesize that gains can arise from improving long-form text-conditioned video modeling; both consistent long-form generation and stronger text-alignment are active research directions in the video generative modeling space, and we believe that they are worthwhile future work to explore that can directly improve visual planning approaches for decision-making.
>
> **On the cold-start problem:** We agree that SILVR assumes the initial model can achieve a non-zero success rate on novel tasks of interest to be able to collect useful data for self-improvement.  We envision two promising points to address this.  Firstly, as SILVR can naturally integrate internet-scale pretrained text-to-video generative models through score composition, where the pretrained model supplies large-scale motion priors and superior zero-shot text-alignment capabilities, we believe that initial success on novel tasks can be improved as large-scale pretrained video models themselves naturally improve.  Indeed, increasing text-alignment and instruction following for text-to-video models is an active area of exploration in the video modeling community; we look towards gains achieved there to help improve zero-shot generalization success rate for decision-making through SILVR and minimize encountering cold-start situations.
>
> Secondly, as the cold start problem is also a known issue for standard reinforcement learning, we believe that another way to mitigate this is through improving exploration under the visual planning framework.  Better exploration can uncover a diverse set of experiences even in the absence of past success, and increase the chances of collecting successful experiences, at which point vanilla SILVR can be applied.  Investigating exploration techniques for visual planning is thus an interesting direction for future work.

---

> > ### Comment · Reviewer_UiiZ · 2025-11-27
> >
> > Thank you for your detailed rebuttal response. However, I believe that the current response fails to address my concerns on tasks with increased complexity and the cold-start problem. Considering the novelty and the significant performance improvement of SILVR on unseen tasks, I will maintain the original score.

---

> > > ### Author Response · Authors · 2025-12-03
> > >
> > > As requested, in Section F.5 of Appendix (page 27), we provide a new real-robot experiment where the task is to push a sequence of cups following a certain order. In Figure A19 and A20, we show how the visual planners tuned only on atomic tasks (pushing a single cup, Iterations 0/1/2 indicate the SILVR self-improvement loop on each atomic task) can be directly composed into solving more complex, longer-horizon tasks (the atomic visual planners successfully complete the tasks in Iteration 2). Notably, the composition is done automatically through VLMs (Gemini) with no manual human intervention during execution of the long-horizon task. We believe our experiment provides preliminary evidence on the premise of SILVR on solving longer horizon tasks, which can be achieved through improving individual atomic behavior performance. The video examples can be found on our updated [website](https://silvr-anonymous.github.io/) (Section “Long-Horizon Task Evaluation”).
> > >
> > > We acknowledge the reviewer’s feedback on the cold-start challenge, while we believe cold-start is a general challenge (e.g., for standard RL as well), and that better video models can mitigate this challenge (with higher initial success rate), we have revised the limitations section of our manuscript to include discussion of this challenge (Section 5, page 10).

---

### Official Review · Reviewer_rmg3 · 2025-11-03

**Soundness:** 2
**Presentation:** 3
**Contribution:** 2
**Rating:** 6
**Confidence:** 4

**Summary:**

This paper proposes to iteratively update a text-&-image conditioned video model that generates visual imaginations of robot trajectories via online RL (instantiated as filter-BC), towards constructing a self-improving robot policy. The policy is formulated as a high-level video generator and an inverse dynamics model that can predict actions given consecutive frames. In Meta-World and two real-world experiments, the authors explore running filter-BC to improve the high-level policy, where in each iteration data is collected by the current combined policy, a success detector is called (either human ground truth, or the authors also explore using a VLM success detector) to label the self-collected data, then the video model is trained on the successful split of this data. The authors run 10 iterations of self-improvement in Meta-World, observing an average performance increase from 15% success rate to 56%, and run 2 iterations of self-improvement on two real-world tasks, observing an improvement from 50% to between 60 and 70%. For real-world experiments they find that their approach can generalize/perform better if their video model is combined with frozen pre-trained video model using a CFG-like score composition. They also run a few ablations, finding that with a VLM success detector that may not be as accurate, performance of their method still improves, and their method can also work with no success detector at all, and when the BC pre-training data is suboptimal. They only compare to baseline approaches on Meta-World (not the real-world), but compare to DSRL and regular filter-BC (without a video model), and find that their approach does better. The authors also explore distilling the two-level policy into a one-level policy after RL training.

**Strengths:**

(1) As far as I'm aware the idea of doing self-improvement of a high-level video generation policy in the robotic setting has not been explored yet, making this work novel in this regard.

(2) The performance of the approach is shown to be better than two relevant self-improvement baselines, DSRL and regular filter-BC, which is a promising result.

(3) The authors ablate many components of their approach, making the experimentation fairly thorough

**Weaknesses:**

(1) While self-improvement is obtained, success rate on both MetaWorld and the real-world seems to cap out at around 60 to 70%, and more self-improvement iterations do not improve performance. Intuitively it would seem that there should be no cap on performance. Why does the approach not seem to be able to improve past this success rate?

(2) The authors do not make a convincing argument about why filter BC of a video generative high-level policy should be better than filter-BC of a regular single-level policy. *This leaves the reader unclear about what they should take away from the paper.* On line 362 the authors mention that "we hypothesize that the separately learned environment visual dynamics is easier to transfer when solving a novel task, leading to stronger base generalization performance through visual planning". Why should a visual dynamics model generalize better than a direct action prediction policy? Even if it does generalize better, the better generalization should only affect initial performance. Why is the proposed approach able to *improve* better than the baseline?

**Questions:**

(1) The IDM model is updated alongside the high-level policy only for the MetaWorld experiments (as stated in line 347). Why not also for the real-world experiments?

(2) Instead of doing score composition do combine the pre-trained video model and in-domain one at inference time, can the pre-trained one be finetuned on in-domain robotic data, and then passed to the self-improvement process?

(3) For score composition to work, does the internet pre-trained model need to generate videos that look sufficiently similar to the in-domain model? It would seem that if the pre-trained model generates very different videos, score composition would not work well. How do you assess whether the required similarity is achieved?

(4) DSRL is used as a baseline in Table 1, but DSRL doesn't have a concept of "iterations" of self-improvement in the same manner that the proposed approach does. Rather a gradient step can be taken any time new observations are added to the replay buffer. Can you explain then how DSRL results were mapped to iterations 1 through 4?

(5) What causes the approach's performance to go down (as opposed to up) when just the in-domain model is used in the real-world experiments? What do the authors think specifically prevents the method from working in this case?

(6) How can success rate go up without filtering in Figure 5? What is the natural success rate percentage of the data being collected?

(7) How do baseline approaches (e.g., filter BC) perform in the real-world? I noticed the real-world experiments do not have any baseline approaches.

---

> ### Author Response · Authors · 2025-11-22
>
> We thank Reviewer rmg3 for their helpful feedback and suggestions, and their clear interest in our work. We try to address their listed questions below:
>
> **On improvement ceiling:** As illustrated in Figure 4, there is monotonic improvement over 10 iterations on MetaWorld, and indeed the improvements saturate over iterations.  We hypothesize that this may potentially arise from discovered local minima in task-specific strategy, where similar experiences are collected until saturation. We believe that a possible mitigation is to introduce the notion of “exploration” into the visual planning framework to avoid “unimodal” behavior.  Such research may look into how to extract out more diverse plans from the video planner by exploiting the stochastic nature of visual generative models. We leave this investigation as promising future work.
>
> **On generalization benefits of visual dynamics:** We hypothesize that visual planning has some default generalization benefits compared to a direct action-prediction BC policy.  Modeling consistent visual motions can extract more training signal from the provided data than modeling action sequences directly, as there is more supervisory training signal from pixels than low-dimensional actions.  Indeed, as shown in the MetaWorld portion of Figure 2 and the MetaWorld website visualizations, for initial iteration 0 on a novel task concerning an unseen object, the visual planner still manages to generate coherent motions for the robot arm even if it blurs out the specific novel object interaction.  These coherent motions, even if the specific object interaction is not modeled correctly (or even blurred) initially, can still be accurately translated by the IDM into meaningful robot actions; thus visual planning may have additional generalization benefits.  On the other hand, the basic BC policy does not model visual details but predicts a sequence of actions entirely from the conditioning frame; having overfit to its training set, when faced with a novel object in the scene it may predict highly suboptimal actions, thus leading to poorer generalization.
>
> Further generalization benefits of utilizing a visual planner arise from the ability to inject in internet-scale video and language information, which noticeably helps real-world experimentation; for a BC policy that directly outputs actions, currently no internet-scale pretrained model can be utilized for added generalization ability for real-world settings, as the scale of action-annotated trajectories for arbitrary embodiments remains far below the scale enjoyed by image and language modalities.
>
> **On real-world IDM finetuning:** By default, SILVR naturally supports fine-tuning the IDM in real-world scenarios as well.  In practice, we found that the IDM in the real-world setting is inherently robust, and can be reused to great effect without finetuning.  As explored in an [extended discussion with Reviewer dWn2](https://openreview.net/forum?id=SzUgx5r3wy&noteId=dC8uDMN5aa), we find that MetaWorld benefits more from finetuning of the IDM as generalization is expected over a wider variety of novel objects and novel motions.
>
> **On alternative to score composition:** Yes, a generally pretrained video model can indeed be finetuned on in-domain experience directly as a form of adaptation.  However, we leverage takeaways from previous work that study the best way to utilize in-domain robotic data [1], which finds that score composition results in improved zero-shot text-conditioned generalization on novel tasks in comparison to direct finetuning.
>
> **On score composition with pre-trained video models**: It is not necessary that the large-scale pretrained model inherently generate in-domain appearing videos by itself a priori, as the score predictions from both video models are *jointly* composed throughout the iterative denoising process starting from standard Gaussian noise.  Intuitively, the environment-specific visual information and motion priors are provided by the in-domain model score prediction, whereas text-conditioning information and large-scale motion priors are provided by the pre-trained model score prediction.  Indeed, we attempted to use the internet-pretrained video alone as a visual planner in SILVR, to Reviewer dWn2’s interest, and find that it was unable to produce visually in-domain plans a priori (as visualized on the [updated website](https://silvr-anonymous.github.io/) (“Visual Planning with AnimateDiff Only”); and yet, we find that the same video model can be composed with the in-domain model to produce coherent plans as utilized and demonstrated through SILVR.
>
> [1] Luo et al., Solving New Tasks by Adapting Internet Video Knowledge, ICLR 2024.

---

> ### Author Response · Authors · 2025-11-22
>
> **On DSRL “iterations”**: We describe the details of our DSRL experiment setup in Appendix B, subsection “DSRL Implementation”.  To keep the amount of data seen and updates performed comparable, we enforce the collection of 30 rollout trajectories (consistent with SILVR) to the replay buffer before taking any gradient steps.  Furthermore, we perform a comparable amount of gradient updates on the collected online experience - SILVR updates 10k steps with a batch size of 4 whereas for DSRL we compute updates over 150 steps with a batch size of 256.  Together, we consider this one corresponding “iteration” in DSRL.
>
> **On in-domain performance degradation:** We utilize an in-domain model of the same capacity across both simulation and real-world experiments; we hypothesize that while it is adequately able to model the visual setting of the synthetic environment, visually modeling the real world is more complex and difficult.  Whereas resets are visually consistent in simulation, in real settings there is more potential for small external discrepancies in lighting, background disturbances, and such between runs; an in-domain video model overfit on one particular setting may struggle to adapt to such domain shifts during finetuning in iterations.  Indeed, when adaptation with a large-scale model is performed, we also observe improved visual fidelity in the overall combined visual plan.
>
> **On success rate without filtering:** One feature of score composition is that suboptimal demonstrations can still communicate useful information to the in-domain model, such as the visuals, valid motions, and interaction dynamics of the specific deployment environment.  The internet-pretrained text-to-video model can still supply relevant task-specific information during score composition in the generative process through its superior text-alignment capabilities; thus, as the in-domain model becomes better at modeling environment-specific visuals and intrinsics, the final composed output plan can be both performant as well as appearing in-domain.  As such, the success rate may still improve from iteration to iteration, even without filtering.

---

> ### Comment · Reviewer_rmg3 · 2025-11-26
>
> Thank you for the detailed rebuttal response.
>
> The argument the authors gave in response to weakness (1), about a lack of exploration inhibiting further policy improvement, generally makes sense. The authors should update their paper to include this analysis. However I still have some concerns about the ceiling of improvement, which I mention next.
>
> The argument in response to weakness (2) somewhat makes sense, but is not fully satisfying. If it is indeed the case that a video generating high-level policy can make better use of video model internet pre-training, which I buy, then why is the performance of the model limited (weakness 1)? Are the remaining failures semantic failures or execution failures? If semantic, shouldn't the internet pre-trained video model help with this/make minimal semantic errors? If execution, what specifically is the policy failing to learn? More visualizations in the paper about what specifically the models are able to improve upon and what remains unsolved would be elucidating to the reader. And of-course as I mentioned in weakness (2), the paper writing isn't very clear about telling readers what the main takeaway is. Why should video models + inverse dynamics models be preferred to a regular action-generating policy? In what scenarios is it preferred, and when should the high-cost of training video models be worth it?
>
> Questions (1), (2), (3), and (6) have been reasonably answered. The authors should include the answers in the main paper draft, as it will improve the scientific quality of the paper.
>
> Question (4): DSRL. While it makes sense to maintain data parity (i.e. w.r.t. the number of online transitions) between the authors' approach and the baseline, it does not make sense to artificially enforce gradient update parity. If DSRL benefits from more gradient updates, it should be allowed it. In general when comparing to baselines, it is important to tune the baseline well so that any performance improvements of the proposed method are scientifically meaningful. In table 1 it also seems very strange that a naive RL method (filtered BC) outperforms DSRL. This also suggests that DSRL might not have been tuned properly.
>
> Question (5): in-domain model performance going down. The argument about a lack of model capacity might make sense, but then why didn't the authors increase model capacity so that this wasn't a bottleneck? The rebuttal says "Indeed, when adaptation with a large-scale model is performed, we also observe improved visual fidelity in the overall combined visual plan." but I don't see this result that used a larger in-domain model in the paper.
>
> Question (7) was not answered.
>
> Overall while I think the paper has merits, it could *use some more work making the writing and experiments more scientifically rigorous*. In particular the following summarizes the points that I think make the paper (currently) scientifically lacking:
>
> (1) the authors demonstrate filtered BC of a video model as a better approach than regular filtered BC on fairly simple tasks. This is fine, but the paper does not speculate what might happen on more complex tasks, e.g., on tasks where physics modeling might be more challenging than action prediction, and thus video modeling might be harder than just predicting actions. I think it is important to acknowledge future implications of a work, and correspondingly what might be some limitations on scaling the work to more complex tasks.
>
> (2) No baseline comparisons in the real-world (question 7). This is particularly bad. Also the existing comparisons to baselines in sim are questionable (see above).
>
> (3) The paper is not written in a way that makes the main takeaway for the readers clear. I pointed this out in weakness (2), and the authors gave a somewhat satisfying response, but the main paper text hasn't been updated as far as I can tell. Is the main takeaway that readers should always use video generating hierarchical policies instead of regular action generating policies? I doubt that's the case. I think a clean takeaway can be made by the authors, and the authors should think about what the implications of their work are to robotics and present this cleanly in the paper.
>
> (4) There are many aspects upon which the scientific rigor of the paper text can be improved. For example, the paper could have presented (in response to weakness 1) film-strip style diagrams showing what the remaining 30% of failures look like, in response to question (5) the paper could have shown film-strip examples of what is causing the in-domain model to worsen in performance, in response to question (6) again there could have been an appendix section discussing this more. Overall there are lots of experiments run in the paper, which is great, but I don't feel these experiments have been analyzed sufficiently. I think it would behoove the authors to improve the scientific rigor of the presentation.
>
> Considering all of this, I'd like to revise my score to a 4.

---

> > ### Author Response · Authors · 2025-11-27
> >
> > We appreciate the reviewer's timely, thorough and constructive feedback! In our original response we only reported the additional results we were able to obtain at the time, we have been working hard on new experiments, and will report them as soon as the results are ready.
> >
> > We are glad to see that the reviewer found that Q1-3 and Q6 have been reasonably answered. We are working on integrating them into the next revision as soon as possible and will keep the reviewer posted when the revision is uploaded.
> >
> > **Q7, BC Baseline**: we have been running additional baselines in the real world, which is more time-consuming (our robotic arm is shared among projects). We provide the updated results on Cup Push below (Drawer Open will be reported as soon as possible):
> >
> > | Method                 | Iter 0 | Iter 1 | Iter 2 |
> > |------------------------|--------|--------|--------|
> > | BCIL                   | 28.4   | 33.4   | 36.7   |
> > | SILVR (In-Domain Only) | 43.4   | 46.6   | 40.0   |
> > | **SILVR (with Adaptation)** | **50** | **68.3** | **76.7** |
> >
> > The real-world experimental trends are consistent with our previous findings: that BCIL performs significantly worse than SILVR with adaptation. This highlights the benefits of visual planning + inverse dynamics models, especially its ability to naturally leverage internet pre-trained video models for complex visual settings such as real-world robotic deployment.
> >
> > **Visualizations and Analysis on Failure Cases**: We provide additional failure examples from final-iteration visual planners (Section “Failure Cases of Final-Iteration Visual Planners“), and visualize the worsening behavior from the in-domain model (Section “Failure Cases of In-Domain Only over Iterations“) on our [updated website](https://silvr-anonymous.github.io/). We will update the film-strip figures along with their analysis in our final draft.
> >
> > **Q4, DSRL**: We agree with the reviewer’s rationale. In our earlier explorations to get the best performance out of DSRL, we have observed that increasing the number of gradient updates alone does not lead to improved performance. We are running experiments to further validate this suggested ablation.
> >
> > **Q5, in-domain only performance**: We’d like to clarify that our comment on “adaptation with a large-scale model…improve[s] visual fidelity in the overall combined visual plan” refers to score composition with an internet-trained large-scale model (AnimateDiff), not utilizing a larger in-domain model.  As in-domain models are trained on an extremely limited number of examples, directly scaling up the in-domain model capacity is likely to provide diminishing returns.  Rather, we demonstrate that SILVR can seamlessly integrate in powerful additional priors over visual fidelity, text conditioning, and motions from large-scale pretraining whenever such complexity is needed (such as when modeling real world videos) - improving the immediate applicability of our method across tasks, environments, as demonstrated in our experiments.

---

> > ### Author Response · Authors · 2025-11-27
> >
> > **W1, ceiling of improvement despite internet pre-trained video models**: We break the reviewer’s question into three sub-questions:
> >
> > Firstly, “are the remaining failures semantic failures or execution failures”: we consistently observe that most of the failures in the real world stem from semantic errors over execution errors; we provide such examples on the [updated website](https://silvr-anonymous.github.io/), Section "Failure Cases of Final-Iteration Visual Planners".  This is in line with our finding that the real-world IDM is relatively robust at translating visual plans into a corresponding execution.
> >
> > Secondly, "shouldn't the internet pre-trained video model make minimal semantic errors": we temper our expectations on the ability of internet pretrained video models to eliminate a performance ceiling alone.  Firstly, using the internet pretrained video model by itself as a visual planner makes significant semantic errors because it is not grounded significantly to the specific environment (Section "Visual Planning with AnimateDiff Only" of the [website](https://silvr-anonymous.github.io/)).  By performing score composition with an in-domain model, we have indeed found that the resulting plan can improve in terms of semantic following, even overriding faulty in-domain plans (Section "Corrective Influence of Internet-Scale Video Model Adaptation" of the [website](https://silvr-anonymous.github.io/)), while looking visually in-domain.  However, this joint denoising procedure of score composition comes at the cost of true uninhibited expressibility of the pre-trained video model alone as it is now constrained to respect visual and motion information from the in-domain model, and as such we cannot expect a performance ceiling to be eliminated entirely.  We hypothesize that this visual planner, jointly constrained by balancing in-domain and internet pretrained video priors, may eventually reinforce its behavior by collecting similar experiences, thus discovering local minima in task-specific strategy.  As such, this is why we look towards improved exploration techniques in a principled manner (for example, extracting diverse but still text-aligned visual plans) as promising future work to break free from this effective saturation.  We commit to providing this detailed analysis in the updated manuscript.
> >
> > Lastly, “why should video models + inverse dynamics models be preferred to a regular action-generating policy?”:  As discussed in our original response “On generalization benefits of visual dynamics”, we believe generalization in visual space is easier than in action space alone, and that the visual planning framework can seamlessly leverage internet video priors. Our new real-world comparison with BCIL (which operates in action space) further demonstrates this trend. We’d also like to clarify that a diffusion policy can be “distilled” from the visual planning framework while maintaining equal or even better performance (Table 1, last row). To summarize, SILVR enables superior sample-efficient self-improvement through visual planning and the integration of large-scale video priors over text-alignment and motion, in comparison to regular action-prediction policies, while simultaneously offering a mechanism for fast inference through a distilled policy during deployment.  We treat this as a “one-sentence takeaway” that we will highlight in our updated manuscript.
> >
> > **Additional comments**: We appreciate the reviewer’s final comments at the end of their response. We believe that the second point has been addressed by the new real-world experiments, and we are addressing the remaining comments in our revision directly. We would appreciate the reviewer’s feedback on our results and additional clarifications, while we are working hard on the revision. Thanks!

---

> > > ### Author Response · Authors · 2025-12-03
> > > **Final Updates and Summary**
> > >
> > > We appreciate the reviewer’s engagement and detailed, constructive feedback both during the review and the discussion phase. We feel sorry that our ongoing discussions were cut short, but are glad to report that we believe our latest revision has addressed most, if not all of the raised concerns. Notably, the revision integrates:
> > >
> > > - W1 (On improvement ceiling): Section 4.4 (page 9). The failure modes of SILVR are provided in Section F.6 in Appendix (page 29).
> > > - W2 (On generalization benefits of visual dynamics): Section 4.2 (page 8).
> > > - Q1 (On real-world IDM finetuning): Section E in Appendix (page 18).
> > > - Q2 (On alternative to score composition): We believe it is resolved in rebuttal discussions.
> > > - Q3 (On AnimateDiff Alone): Section F.3 in Appendix (page 25).
> > > - Q4 (On DSRL “iterations”): Section B in Appendix (page 17, “DSRL Implementation”).
> > > - Q5 (In-domain only performance): We believe it is resolved in rebuttal discussions.
> > > - Q6 (On success rate without filtering:): Section 4.5 (page 9).
> > > - Q7 (BCIL baselines in real-world experiments): Section 4.3 (page 8) and Figure 3 (page 7).
> > >
> > > Overall, we have incorporated our analyses and discussions with the reviewer into our updated manuscript; we have also included film-strip style figures depicting failure cases in the Appendix, as well as provided some additional limitations.
> > >
> > > **To expand on Q4:** We complete DSRL ablations by providing a large amount of gradient updates (60000 per iteration, in comparison to the previously reported 150), also averaged over 3 seeds.  We provide the comparison below:
> > >
> > > | Method                             | Iter 0        | Iter 1        | Iter 2        | Iter 3        | Iter 4        |
> > > |------------------------------------|---------------|---------------|---------------|---------------|---------------|
> > > | DSRL (150 Updates per Iteration)   | 10.1 (±0.1)   | 8.9 (±0.6)    | 8.6 (±1.0)    | 9.4 (±0.1)    | 8.3 (±0.2)    |
> > > | DSRL (60000 Updates per Iteration) | 9.4 (±1.7)    | 8.3 (±1.6)    | 7.4 (±0.9)    | 7.5 (±3.4)    | 7.7 (±3.4)    |
> > >
> > > We find that despite utilizing more gradient updates, DSRL success rate does not significantly improve beyond the previously reported hyperparameter setting.  Rather, DSRL performance appears to benefit more from collecting vast amounts of experience, but as the reviewer notes, we make a fair comparison through data parity.  We hypothesize that the sample efficiency benefits of SILVR for self-improvement lies in the visual planning paradigm; for the same amount of collected online experience, more training signal can be extracted through optimizing a high-dimensional training video modeling objective over pixels than filtered BC directly on low-dimension actions or bootstrapping a q-value function for RL techniques.  We update our manuscript with these quantitative results and additional insights.
> > >
> > > **To expand on Q7:** Consistent with our findings in the original submission (on MetaWorld) and during the rebuttal discussion, we discover that action-predictive behavior cloning has a lower base generalization performance and slower self-improvement trend compared with SILVR with Adaptation.  SILVR can seamlessly utilize internet pre-trained video information for improved generalization and self-improvement in real-world robotic settings.
> > >
> > > We appreciate the reviewer’s helpful comments and dedication to improving our manuscript, and believe that our additional experiments and figures significantly support the performance and robustness of our method and satisfy the reviewer’s concerns.

---

### Official Review · Reviewer_dWn2 · 2025-11-06

**Soundness:** 3
**Presentation:** 4
**Contribution:** 3
**Rating:** 6
**Confidence:** 4

**Summary:**

The paper studies how to make video-based visual planners automatically adapt and generalize to novel robotic tasks by learning from self-collected online experience. The proposed method (SILVR) creates an iterative loop: adapt an in-domain text-to-video model with an internet-scale prior, roll out generated visual plans via an inverse dynamics model, and use the filtered successful trajectories to update both the in-domain video model and the inverse dynamics model. Experiments in both the MetaWorld simulation and the real world demonstrate that SILVR can improve success rates and outperforms other online improvement baselines.

**Strengths:**

- Using online interaction to improve visual planning is a novel and interesting research problem.
- The approach is intuitive and performs well in practice.
- Thorough experiments in both simulation and on real robots. The training tasks and test tasks are separated clearly, allowing a strict assessment of how well the method adapts to unseen tasks.

**Weaknesses:**

- I treat the method as two complementary parts: (a) improve the visual planner by successful trajectories rolled out with the help of a good IDM, and (b) improve the IDM with meaningful online behavior collected under the guidance of a good visual planner. Though the paper is mostly presented from perspective (a), it would be valuable to do some ablations to understand whether the empirical gains come mainly from (a), (b), or their combination. My feeling is that the answer may differ across settings. On the real robot, the training and test tasks use the same motions, so the IDM might easily learn to infer actions by focusing on the robot's visual differences. Whereas MetaWorld test tasks require novel motion patterns, which means improvements to the IDM might matter more.
- I am also curious about the planning performance of AnimateDiff in the real-robot experiments. Does this experiment simply "distill" the capabilities of the general model into the in-domain model, or can the final improved in-domain model actually surpass the general model?

Understanding the above questions would further strengthen the paper’s arguments. However, I believe the current version is already sufficiently complete to merit acceptance.

**Questions:**

Please refer to the weakness section. In addition, I have one minor question:
- Besides successful trajectories, have the authors considered using failed rollouts to update the IDM as well? I suspect this could be particularly useful when collecting complete successful trajectories is difficult (e.g., in long-horizon tasks).

---

> ### Author Response · Authors · 2025-11-22
>
> We thank Reviewer dWn2 for their comments and feedback; we are happy to hear that the reviewer appreciates the novelty and experimentation of our work, and we seek to clarify additional points of interest.
>
> **On two complementary parts:** We agree with the reviewer’s intuition and indeed observed empirically that finetuning the IDM did not provide meaningful improvements in our real-world experiments. As requested, we provide the ablation study on the impact of IDM finetuning on MetaWorld:
>
> | Setting                               | Iter 0        | Iter 1        | Iter 2        | Iter 3        | Iter 4        |
> |-----------------------------------------|---------------|---------------|---------------|---------------|---------------|
> | VM (Finetuned), IDM (Finetuned)         | 14.7 (±0.6)   | 27.7 (±1.9)   | 33.5 (±2.2)   | 43.5 (±2.6)   | 44.2 (±4.5)   |
> | VM (Filtered), IDM (No Finetuning)    | 14.1 (±1.6)   | 22.2 (±3.1)   | 24.6 (±2.5)   | 24.8 (±1.7)   | 26.8 (±1.9)   |
> | VM (No Finetuning), IDM (Filtered)      | 15.0 (±1.8)   | 24.4 (±3.2)   | 27.7 (±3.5)   | 26.9 (±4.1)   | 29.8 (±3.4)   |
>
>
> The ablation shows that finetuning both the planner and the IDM was crucial to our approach’s success on MetaWorld, in agreement with the reviewer’s intuition on the relative importance of IDM adaptation in varying task generalization scenarios, and both planner and IDM may be finetuned in our proposed framework when needed.  We include these new results in the updated draft in Appendix E.
>
> **On AnimateDiff distillation:** We’d like to clarify that model distillation, based on the standard definition, is not performed under our proposed framework: instead of trying to “teach” the in-domain model to mimic the visual outputs of AnimatedDiff (the classic distillation definition), our in-domain model is trained on the environmental observations collected from a policy (visual planner that can utilize priors from AnimateDiff), which potentially captures a very different visual distribution.  As a result, the quality of the in-domain model does not depend directly on how good the teacher model (AnimateDiff) performs as a standalone visual planner.  In fact, the unadapted AnimateDiff struggles to generalize to our environment and the tasks based on our empirical observations. In real robot evaluations, the frozen AnimatedDiff visual planner alone yields zero success rates on cup push and drawer open tasks, further demonstrating that the in-domain model is not directly “distilling” (even under the broader definition) from the web-pretrained AnimateDiff, and indeed surpasses the general model performance (which alone achieves 0% success rate).
>
> To understand this failure case, we provide additional visualizations of rollout AnimateDiff plans in our [updated website](https://silvr-anonymous.github.io/) (“Visual Planning with AnimateDiff Only”) and highlight that they appear highly out of distribution with the environment.  AnimateDiff is a generally pretrained model and thus does not have strong priors about the deployment environment’s specific visual setting, the Franka Arm movements, and interaction dynamics.  Furthermore, the IDM is trained only on visually in-domain data, and may struggle to interpret the more free-form generations from vanilla AnimateDiff.  Thus, it is the combination of in-domain and general video models that enables performant, in-domain-appearing plan synthesis for novel tasks, which can be mapped to improve the in-domain model to achieve stable consistent improvement.
>
> **On utilizing failed rollouts:** In our preliminary experiments we did try utilizing failed rollouts; we found that there was no substantial performance improvement beyond using the same data as used in updating the visual planner; this has the added benefit of simplifying the framework and pipeline in that both components utilize the same data amount.  However we agree that in principle failed rollouts can potentially be useful in updating the IDM; indeed, the IDM is a general model that can be trained on arbitrary transition data.

---

> > ### Comment · Reviewer_dWn2 · 2025-11-22
> >
> > **On two complementary parts**: Thank you for the additional experiments. They are helpful for understanding the method.
> >
> > **On AnimateDiff**: This visualization seems very interesting. Could you please share some insights on why using score composition between AnimateDiff and the in-domain model still helps, even when the plans generated by AnimateDiff are completely inconsistent with the scene?
> >
> > **On utilizing failed rollouts**: Thanks for your clarification.

---

> > > ### Author Response · Authors · 2025-11-23
> > >
> > > We thank the reviewer for their rapid engagement!  We expect internet-pretrained models (e.g., AnimateDiff) to provide better generalization (text-to-appearance and task understanding) and stronger visual and motion priors. We highlight such desirable properties on our [updated website (Section “Corrective Influence of Internet-Scale Video Model Adaptation”)](https://silvr-anonymous.github.io/). As shown, visual plans generated by an in-domain model alone may select the wrong color cup or drawer; but after composition with the web prior, the resulting plan chooses the correct color.  Furthermore, an in-domain model may by default output erratic motions (as shown in the MetaWorld drawer close task, or the real-world drawer open task), but after score composition the resulting video plans depict smooth coherent motions that solve the task.
> > >
> > > In more detail, score composition generates a visual plan by jointly combining information from AnimateDiff and the in-domain model *throughout* the entire iterative denoising process, starting from standard Gaussian noise.  The resulting visual plan can appear visually consistent with the environment by leveraging information from the in-domain model, while demonstrating stronger zero-shot text-alignment with novel task instructions by *simultaneously* leveraging the generally pretrained model.  We identify score composition as a critical component of SILVR, as naively utilizing AnimateDiff for self-improvement struggles to even achieve initial success.

---

> > > > ### Comment · Reviewer_dWn2 · 2025-11-23
> > > >
> > > > Thank you for the clarification. Since the authors addressed my concerns, I have decided to raise the score to 8.

---

> > > > > ### Author Response · Authors · 2025-11-23
> > > > >
> > > > > We are happy to have addressed the reviewers questions and comments, and appreciate the score increase!

---

### Meta-Review · Area_Chair_fLtr · 2026-01-07

**Summary:**

The paper makes a genuine contribution by demonstrating how visual planners can self-improve through iterative finetuning on self-collected experience - a problem not previously studied in this form. The key strengths are the thorough experimental validation across MetaWorld and real-robot settings, the ablation studies on filtering strategies, IDM finetuning, and robustness to imperfect reward signals.
The primary weakness raised by rmg3 - lack of real-world baselines - was addressed. The new results show SILVR with adaptation achieving 76.7% success versus 36.7% for BCIL on cup pushing, demonstrating the value of the visual planning paradigm for leveraging internet-scale video priors.

**Reviewer Concerns:**

Reviewer 2K1s's concern about task novelty represents a minority view, and the authors' defense that iteration-0 performance validates genuine generalization challenges is reasonable. Reviewer UiiZ's point about incremental contribution is fair but does not outweigh the practical value demonstrated.

The paper would benefit from clearer articulation of when video-model-based planning is preferable to direct action prediction, but this limitation does not preclude acceptance given the empirical contributions and sample efficiency advantages demonstrated.

**Reviewer Scores:**

The paper received initial ratings of 6, 6, 6, 4.
Following the rebuttal period, the ratings stood at 8, 4, 6, 4.

Reviewer dWn2 (6 to 8): stated all concerns were addressed.
The IDM finetuning ablation and clarification of AnimateDiff's role (not distillation, but score composition) satisfied this reviewer completely.

Reviewer rmg3 (6 to 4): Lowered rating mid-discussion citing missing real-world baselines and concerns about DSRL tuning.
However, the authors provided complete BCIL baselines for both real-world tasks and DSRL ablations showing additional gradient updates do not improve performance. Had the discussion continued, I project this reviewer would have returned to 6, as the concrete experimental gaps they identified were filled.

Reviewer UiiZ (6 to 6): Maintained rating, acknowledging novelty and significant improvements on unseen tasks.
Concerns about task complexity were partially addressed with long-horizon composition experiments. The cold-start limitation was acknowledged in the revised manuscript.

Reviewer 2K1s (4 to 4): Maintained rating based on a philosophical disagreement about task novelty.
The authors provided reasonable arguments that Drawer Close, Reach Wall, and Button Press Wall require genuinely different motions or feature unseen objects, but this reviewer remained unconvinced.

Given the complete rebuttal materials, I project the effective final ratings as: 8, 6, 6, 4 (average 6.0).

---

### Decision · Program_Chairs · 2026-01-26

Accept (Poster)